# Uncovering the small proteome of *Methanosarcina mazei* using Ribo-seq and peptidomics under different nitrogen conditions

Muhammad Aammar Tufail [1], Britta Jordan[1], Lydia Hadjeras[2], Rick Gelhausen [3], Liam Cassidy[4], Tim Habenicht [1], Miriam Gutt [1], Lisa Hellwig[1], Rolf Backofen [3], Andreas Tholey [4], Cynthia M. Sharma [2] & Ruth A. Schmitz [1] ✉

The mesophilic methanogenic archaeal model organism *Methanosarcina mazei* strain Gö1 is crucial for climate and environmental research due to its ability to produce methane. Here, we establish a Ribo-seq protocol for *M. mazei* strain Gö1 under two growth conditions (nitrogen sufficiency and limitation). The translation of 93 previously annotated and 314 unannotated small ORFs, coding for proteins ≤ 70 amino acids, is predicted with high confidence based on Ribo-seq data. LC-MS analysis validates the translation for 62 annotated small ORFs and 26 unannotated small ORFs. Epitope tagging followed by immunoblotting analysis confirms the translation of 13 out of 16 selected unannotated small ORFs. A comprehensive differential transcription and translation analysis reveals that 29 of 314 unannotated small ORFs are differentially regulated in response to nitrogen availability at the transcriptional and 49 at the translational level. A high number of reported small RNAs are emerging as dual-function RNAs, including sRNA$_{154}$, the central regulatory small RNA of nitrogen metabolism. Several unannotated small ORFs are conserved in *Methanosarcina* species and overproducing several (small ORF encoded) small proteins suggests key physiological functions. Overall, the comprehensive analysis opens an avenue to elucidate the function(s) of multitudinous small proteins and dual-function RNAs in *M. mazei*.

An unexpected complexity and density of genes within microbial genomes have been revealed by the steadily growing number of sequenced prokaryotic genomes in combination with high throughput OMICS profiling technologies such as next generation sequencing of DNA and RNA and optimized proteomics[1,2]. In addition, discovering genes encoding longer proteins or non-coding RNAs, genome-wide studies have also revealed a potential wealth of small proteins in all kingdoms of life. Small proteins are defined here as ribosomal synthesized proteins of ≤ 70 amino acids (aa) in length that are translated from small open reading frames (sORFs). They have been frequently overlooked in the past due to various technical and methodological difficulties e.g., in mass spectrometry[3,4]. Recent emerging tools and

[1]Institute for General Microbiology, Kiel University, 24118 Kiel, Germany. [2]Institute of Molecular Infection Biology, University of Würzburg, 97080 Würzburg, Germany. [3]Bioinformatics Group, Department of Computer Science, University of Freiburg, 79110 Freiburg, Germany. [4]Systematic Proteome Research & Bioanalytics, Institute for Experimental Medicine, Kiel University, 24105 Kiel, Germany. ✉e-mail: rschmitz@ifam.uni-kiel.de

evidences demonstrated that small proteins in eukarya, bacteria, and viruses are implicated in important and diverse cellular functions, such as transport, sporulation, signal transduction, virulence, symbiosis or antiCRISPR activity[4–12].

Small proteins in the domain of archaea are underrepresented in recent studies, and only few small proteins are characterized until now[13,14]. This occurs to the smaller number of identified and cultivable archaea. Many archaea live in extreme habitats or in symbiosis with multicellular organisms. Consequently, due to the challenging experimental work and prediction tools optimized for bacteria, less is known about small proteins in archaea[14]. *Methanosarcina mazei* strain Gö1 is an archaeal model organism and represents a methylotrophic methanogenic archaeon[15]. It produces methane from carbon sources like $CO_2$ plus $H_2$, methanol, methylamine or acetate[16–19] and can fix molecular nitrogen under nitrogen (N) limitation[20]. The molecular mechanisms of regulating nitrogen fixation in response to environmental conditions is well studied on transcriptional and post-transcriptional level[21–27] and the established genetic system allows to perform functional studies in vivo[28–30]. Using a differential RNA-seq approach to identify regulatory small (s)RNAs of *M. mazei* under N sufficient ( + N) and N limited (-N) condition, 44 sRNAs encoding for a putative small protein were already discovered in 2009[27]. Following genome-wide RNAseq studies under different growth conditions analysed by manual inspection or automated bioinformatics tools predicted a total of 1442 sRNAs encoding for small proteins ≤ 70 aa[14]. The first three identified small proteins of *M. mazei* were experimentally verified and quantified by LC-MS in 2015, demonstrating an increased amount of the small proteins sP36 and sP41 in the mid exponential phase under -N[31]. The first functional characterized small proteins from *M. mazei* are, sP26 and sp36. sP26 has been shown to interact with glutamine synthetase ($GlnA_1$) to stabilize its dodecameric structure under -N resulting in increased activity[32]; sP36 is required for ammonium transporter ($AmtB_1$) regulation[33].

RNA-seq technologies have been applied to study prokaryotic transcriptomes and have revolutionized the discovery of novel transcripts, including regulatory sRNAs. While RNA-seq can provide evidence for sORF transcription and can greatly facilitate gene prediction, it cannot be used to directly distinguish between coding and non-coding transcripts, to provide sORF coordinates, or to predict protein abundance since mRNA expression does not necessarily correlate with protein levels due to post-transcriptional regulation. The development of Ribo-seq, which is based on deep-sequencing of ribosome protected footprints to determine genome-wide ribosome occupancy, is more amenable to detection of sORFs, and has provided direct evidence for translation of a wealth of novel unannotated small proteins in diverse organisms[34–37], viruses[38] and more recently also in haloarchaea[39,40]. Consequently, Ribo-seq can be used to investigate whether some of the recently-revealed sRNAs in diverse prokaryotes may in fact be either short mRNAs encoding small proteins or act as dual-function RNAs, having both regulatory and coding potentials[41–43].

In this study, we developed and applied Ribo-seq coupled with RNA-seq on *M. mazei* strain Gö1 to map its global translatome with a particular focus on the small proteins. The use of MNase in our Ribo-seq protocol effectively cleaved mRNA regions that were not shielded by ribosomes, like untranslated regions, enabling distinction between translated and untranslated regions.

Besides detecting the translation of 93 annotated sORFs, some of which are included in recent genome annotation updates, we also discovered 314 translated unannotated sORFs. The translation of 62 annotated and 26 unannotated sORF encoded small proteins was further validated via re-assessment of previous proteomics datasets (both top-down and bottom-up) using a protein database incorporating the unannotated small proteins. Importantly, the re-analysis of top-down datasets further validated many small proteins as being translated as full-length proteoforms, with several identified small proteins

present as proteoforms with and without excision of the first methionine. Further targeted in vivo validation by western blot analysis confirmed translation of 13 out of 16 small proteins, thereby validating predictions derived from Ribo-seq data. Interestingly, differential analysis (-N vs. +N) inferred regulation for some of the identified translated sORFs, strongly suggesting important physiological functions. Overall, this Ribo-seq analysis provides a large catalogue of unannotated *M. mazei* sORFs (314), many of which are conserved in (methano)archaea, serving as a comprehensive resource to further investigate their functions in *M. mazei* e.g., their role in N metabolism or confirm and study their potential role as dual-function sRNAs.

## Results

### Set up of Ribo-seq optimized for *M. mazei*

To generate a translatome map that provides a comprehensive depiction of translated annotated sORFs and to discover unannotated sORFs in *M. mazei* Gö1, we optimized the Ribo-seq protocol initially proposed by[40,44,45] and further developed it to suit the characteristic of this particular methanoarchaeon under two different growth conditions ( + N and -N) (Fig. 1A for more detailed information). The polysome profile analysis confirmed successful capture of translating ribosomes (Fig. 1B, depicted by black lines), followed by conversion of *M. mazei* polysomes[46,47] into monosomes using MNase digestion (Fig. 1B)[48–50]. The effective implementation of Ribo-seq for *M. mazei* was validated by examining the Ribo-seq coverage of translated ORFs and known non-coding transcripts. Exemplarily shown for the protein-coding gene *psmB* (MM_0694; encoding a proteasome subunit), which exhibited greater cDNA read coverage in the Ribo-seq library as compared to the paired RNA-seq library (Fig. 1C, left panel), indicating translation of this gene. Conversely, the sRNA162, a known non-coding regulatory sRNA[51], displayed high read coverage of cDNA in the RNA-seq but no coverage in the Ribo-seq library (Fig. 1C right panel). This observation further validates its non-coding nature. Furthermore, the read coverages of the 5'- and 3'-UTRs of *psmB* demonstrated higher levels in the RNA-seq library compared to the Ribo-seq library (Fig. 1C, left panel). This indicates that mRNA regions that are not translated or unprotected regions were effectively degraded by MNase, supporting the validity of our Ribo-seq approach. Similarly, the operon comprising four small proteins (MM1355-MM1358, TRAM domains) also displayed successful digestion of the 5'- and 3'-UTRs (Fig. 1D). This further confirms the reliability of the Ribo-seq datasets in distinguishing translated regions from non-translated regions.

### Insights into protein translation in *M. mazei* through Ribo-seq

Gene annotation presents various challenges, often associated with potential errors and mis-annotation of genes resulting from reliance on computational methods and experimental validation is necessary to confirm gene annotations and improve genome annotation accuracy[52]. During the manual inspection of our Ribo-seq data, we encountered several ORFs, where the Ribo-seq coverage does not fit the annotated ORF. This discrepancy could be attributed to various factors, including the presence of a potential upstream open reading frames (uORFs), or N-terminal extensions or truncations[46]. For instance, the Ribo-seq coverage of *rpoK* (MM_1759) exhibited a difference from the annotated ORF coverage due to the N-terminal extension (Fig. 2A). In contrast, the Ribo-seq coverage of MM_2572 indicated translation initiation downstream to the annotated start codon, which caused the N-terminal truncation (Fig. 2B). In such cases the blastp analysis can help identify potential N-terminal extensions or truncations by comparing the protein sequence of interest to a database of known protein for *Methanosarcina* spp. (Fig. 2, lower parts). Moreover, Ribo-seq analysis can provide valuable possible existence of unannotated genes, missed from genome annotations. When Ribo-seq based predictions identified an ORF in a genomic region covered by ribosome footprints, and not previously annotated as a protein-coding

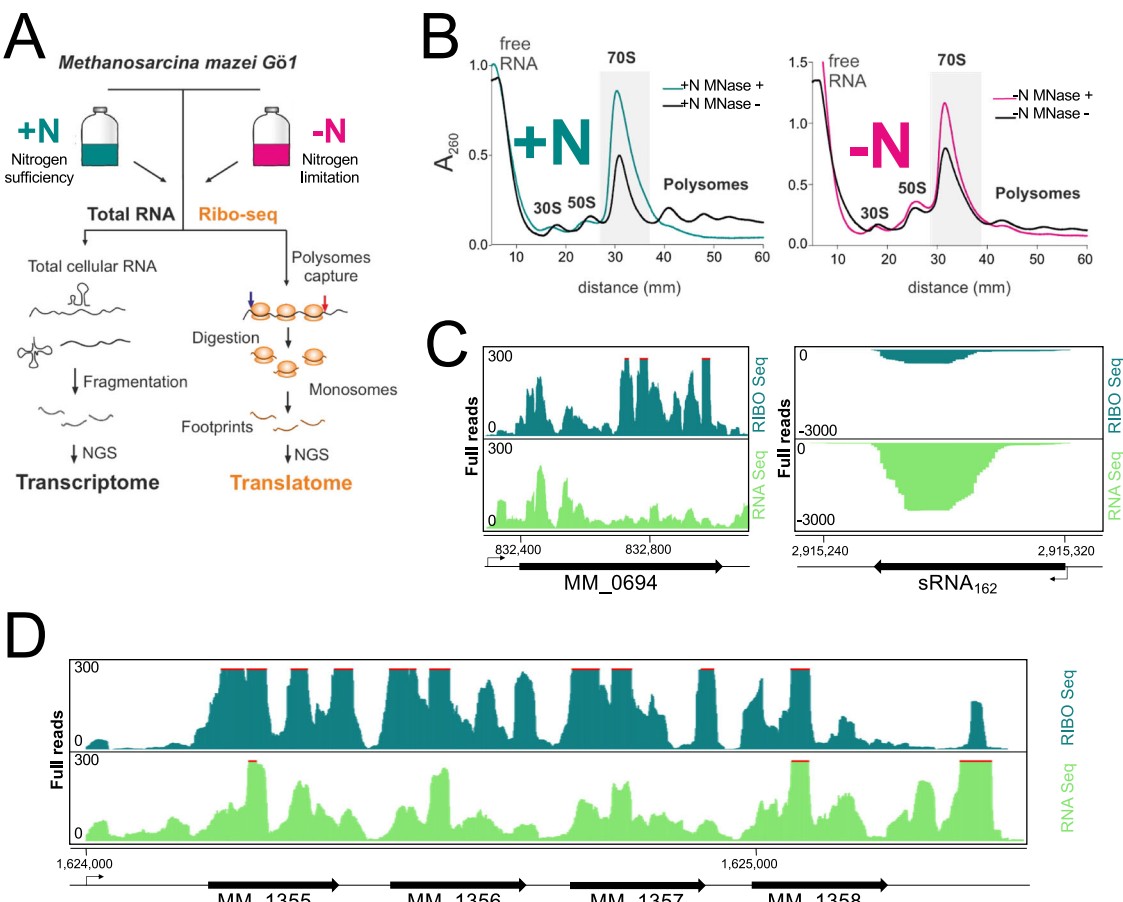

**Fig. 1 | Implementation of ribosome profiling (Ribo-seq) in *M. mazei*.**
**A** Illustrative diagram of Ribo-seq process used to map the translatome of *M. mazei* in two growth conditions: Translating ribosomes were initially detected on the mRNAs through the polysome fraction. MNase digestion eliminated unprotected mRNA regions, leading to the transformation of polysomes into monosomes. The translatome in specific experimental conditions was profiled by subjecting 20-34 nucleotide footprints, which were protected and isolated by 70S ribosomes, to cDNA library preparation and deep sequencing. **B** Fractionation of the cell lysates using sucrose gradients: To avoid the run-off of polysome, cells were harvested during the exponential growth phase using a rapid-chilling method (see Methods). Upon MNase digestion, monosomes were enriched compared to the untreated

sample, as indicated by the 70S peak in the green ( + N) and red (-N) profile, contrasting with the Mock black profile. The monosomes are indicated by a shaded box. Absorbance was measured at a wavelength of 260 nm. **C** JBrowse genome browser screenshots show that our Ribo-seq and RNA-seq datasets distinguish between translated regions (e.g. *psmB* (MM_0694) encoding archaeal proteasome endopeptidase complex subunit beta) and untranslated regions like the non-coding sRNA$_{162}$. **D** The operon of four TRAM domain containing small proteins shows enriched read coverage in the Ribo-seq library along their coding parts but in contrast to the RNA-seq library not in the regions (3′ and 5′ UTRs) un-protected by ribosome. Angled arrows in C and D indicate transcriptional start site (TSS).

gene, it may indicate the existence of an unannotated gene. In our Ribo-seq datasets, we identified 13 such unannotated ORFs (Supplementary Data 1), that were not previously annotated in the annotation version used in this analysis. However, we found that four of these unannotated ORFs are already included in new RefSeq annotation of *M. mazei* strain Gö1 (November 2022). One of the identified unannotated ORFs, named ORF_01 (Supplementary Data 1), is 100% conserved in *M. mazei* using tblastn analysis, but no significant protein homology was found using blastp (Fig. 2C). To confirm the presence of the predicted protein product encoded by the putative gene, and to investigate potential N-terminal extensions or truncations in a protein of interest, additional experiments may be necessary such as mass spectrometry and proteogenomics[53].

### Global translatome of annotated ORFs in *M. mazei* under +N vs. -N

*M. mazei* genome has a total of 3438 annotated ORFs. The DeepRibo algorithm predicted 1566 and 1430 annotated ORFs under +N and -N conditions based on the Ribo-seq data, respectively, overall detecting 47% of all annotated ORFs (Fig. 3A). To detect translation, we determined translation efficiency (TE), which refers to how efficiently a

particular region of an mRNA is translated into protein. This can be calculated as the ratio of Ribo-seq coverage (the number of ribosomes found on a region of mRNA) to total mRNA presence. Consequently, it's a measure of how much of the mRNA that's present is actually being used to generate a protein. Analysis of the start and stop codon distribution of annotated ORFs (Fig. 3B) shows that *M. mazei* favours the canonical start ATG by 79%. The two mainly used stop codons are TAA (53%) and TGA (42%).

In general, by comparing Ribo-seq coverage to a paired RNA-seq library for a given gene, ORF boundaries and 5′/3′ UTRs, if they exist, can be defined and TE (translational efficiency; Ribo-seq/total RNA ratio) can be calculated. We used the following parameters on the Ribo-seq data under +N and -N conditions: TE of ≥ 0.5 and RNA-seq and Ribo-seq Reads Per Kilobase per Million mapped reads (RPKM) of ≥ 10, when prediction by DeepRibo algorithm was needed, a positive DeepRibo score > 0 was chosen for cut-off criteria (Fig. 3A). Global inspection of TE across our dataset for different annotated gene classes (CDS: all annotated coding sequences, sORFs, sRNAs, and tRNAs) showed as expected that protein coding features had a higher mean TE (TE(CDS = 4.49), TE(Annotated sORFs=4.96)) when compared to non-coding sRNAs and tRNAs (TE(sRNAs=1.13),

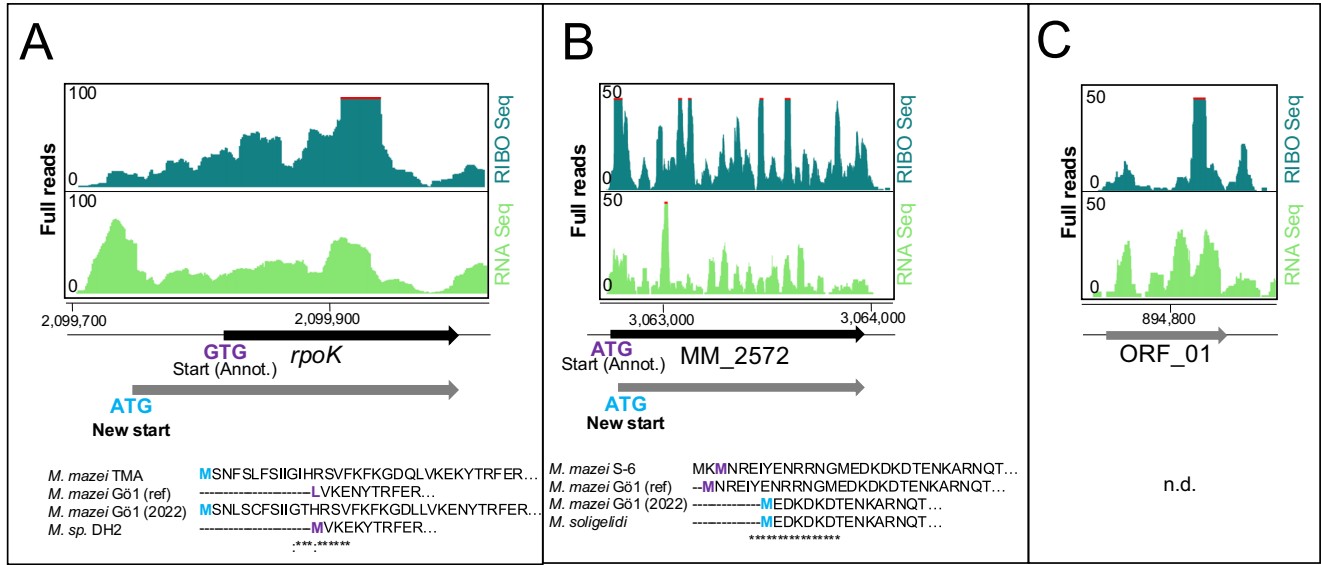

**Fig. 2 | Lessons from Ribo-seq about protein synthesis in *M. mazei*. A** N-terminal extension of *rpoK* gene: Amino acid alignment shows annotated start of translation from different strains. **B** N-terminal truncation of MM_2572: Amino acid alignment based on blastp search shows N-terminus of homologues from different strains. Ref, reference annotation used in this study. **C** An unannotated ORF (ORF_01, Supplementary Data 1) encoding an 80 aa protein, was found during the manual curation of the Ribo-seq predictions with no hits in blastp (n.d.).

TE(tRNAs=0.19)) (Fig. 3C). The comparison of global fold-change values between the -N and +N conditions for RNA-seq and Ribo-seq levels depicted in Fig. 3D, demonstrated a strong positive Pearson's correlation coefficient (r = 0.87). This suggests that the transcriptional regulation of these genes is the primary response to N limitation. However, a subset of genes displayed exclusive significant regulation at either the transcriptional level (depicted by green dots) or the translational level (depicted by blue dots) as shown in Fig. 3D. Specifically, RNA-seq analysis revealed significant regulation exclusively in 148 genes, while the Ribo-seq data indicated exclusive effects in 532 genes, which might be due to the posttranscriptional regulation.

A comprehensive transcriptomics and translatomics analysis provided strong evidence for N dependent regulation in *M. mazei*. Differential gene expression analysis was conducted using the three different tools xtail, riborex and deltaTE for global annotated translatome, small annotated proteome (see The annotated small proteome of *M. mazei* under +N and -N) and the small unannotated proteome (see Manual inspection of unannotated small proteome under +N and -N) under two growth conditions, +N and -N. We used deltaTE outputs because they were more robust and consistent, as they identified the regulation of previously known differentially regulated sORFs such as sORF36 and sORF44[31]. Transcriptomics and translatomics analysis of global annotated proteome with deltaTE revealed the differential regulation of a total of 3774 ORFs. Out of which, 152 annotated ORFs were significantly upregulated (e.g., the *nif* genes) ($p < 0.05$ & log2 Fold Change > 1) and 185 annotated ORFs were significantly downregulated (e.g., *cobN*) ($p < 0.05$ & log2 Fold Change < −1) at RNA level under -N (Fig. 3E and Supplementary Data 2). On the other hand, 247 were significantly upregulated ($p < 0.05$ & log2 Fold Change > 1) and 414 were significantly downregulated (e.g., subunits of RNA polymerase, *rpoB/A*) ($p < 0.05$ & log2 Fold Change < −1) at RIBO level under -N (Fig. 3F and Supplementary Data 2). We performed enrichment analysis (EA) using *clusterProfiler* to identify biological processes associated with differentially expressed genes (DEGs) under -N conditions. Among the three most enriched Gene Ontology (GO) terms for biological processes under -N conditions were 'nitrogen fixation, and 'nitrogen cycle metabolic process' (Fig. 3G). On the other hand, top three biological processes which were downregulated under N stress conditions (-N) include 'biosynthetic processes', 'organic substances biosynthetic

processes', and 'cellular biosynthetic processes' (Fig. 3H). Furthermore, Supplementary Fig. 1 shows the percentage share of each codon by category.

## The annotated small proteome of *M. mazei* under +N and -N

In *M. mazei* 3438 genes are annotated in the used reference annotation[15], (ASM706v1). Of these, 184 are categorized as small proteins with a length of 70 aa or less. These genes were investigated manually for translation following the criteria described in (2.3) and in the Methods section. 93 of the 184 annotated small proteins were labelled as translated in the two tested N conditions (Supplementary Data 3). The length distribution (Fig. 4A) shows that most of the annotated respective small proteins are longer than 60 aa. The smallest annotated sORF encodes for 35 aa. However, only half of the longer annotated sORFs is translated in the tested conditions. The majority of the untranslated annotated sORFs is labelled as hypothetical proteins and annotated based on computational analysis. Further investigation showed that these 93 translated genes favour the canonical start ATG with approximately 83% and the common TAA as their favourite stop codon with 60% (Fig. 4B). This distribution is similar to the start and stop codon usage of genes encoding proteins longer than 70 aa where ATG was used as start codon in 79% of longer genes and 54% of the genes have TAA as stop codon (Fig. 3B). Of all small proteins, 57% are encoded in intergenic regions and 43% are located within operon structures (Fig. 4C).

Transcriptomics and translatomics analysis of 184 annotated sORFs under different N availability revealed intriguing results. Under -N, 5 sORFs were found to be upregulated and 11 sORFs downregulated at the RNA level (as seen in Fig. 4D and Supplementary Data 2). On the other hand, at the RIBO level, 9 sORFs were observed to be upregulated while 22 sORFs were downregulated (depicted in Fig. 4E and Supplementary Data 2), highlighting the complex regulatory mechanisms at work in gene expression. One of the highly upregulated genes is MM_RS13290 (encoding small protein sP36), which has been discovered in 2015[31] and for which we recently demonstrated a crucial function in the post translational regulation of the ammonium transporter AmtB₁[33]. The riboprofiling of sORF36 is visualized in Fig. 4G.

The 70 aa long MoaD protein is important for the Molybdenum pterin cofactor synthesis (cofactor of some of the metabolic enzymes),

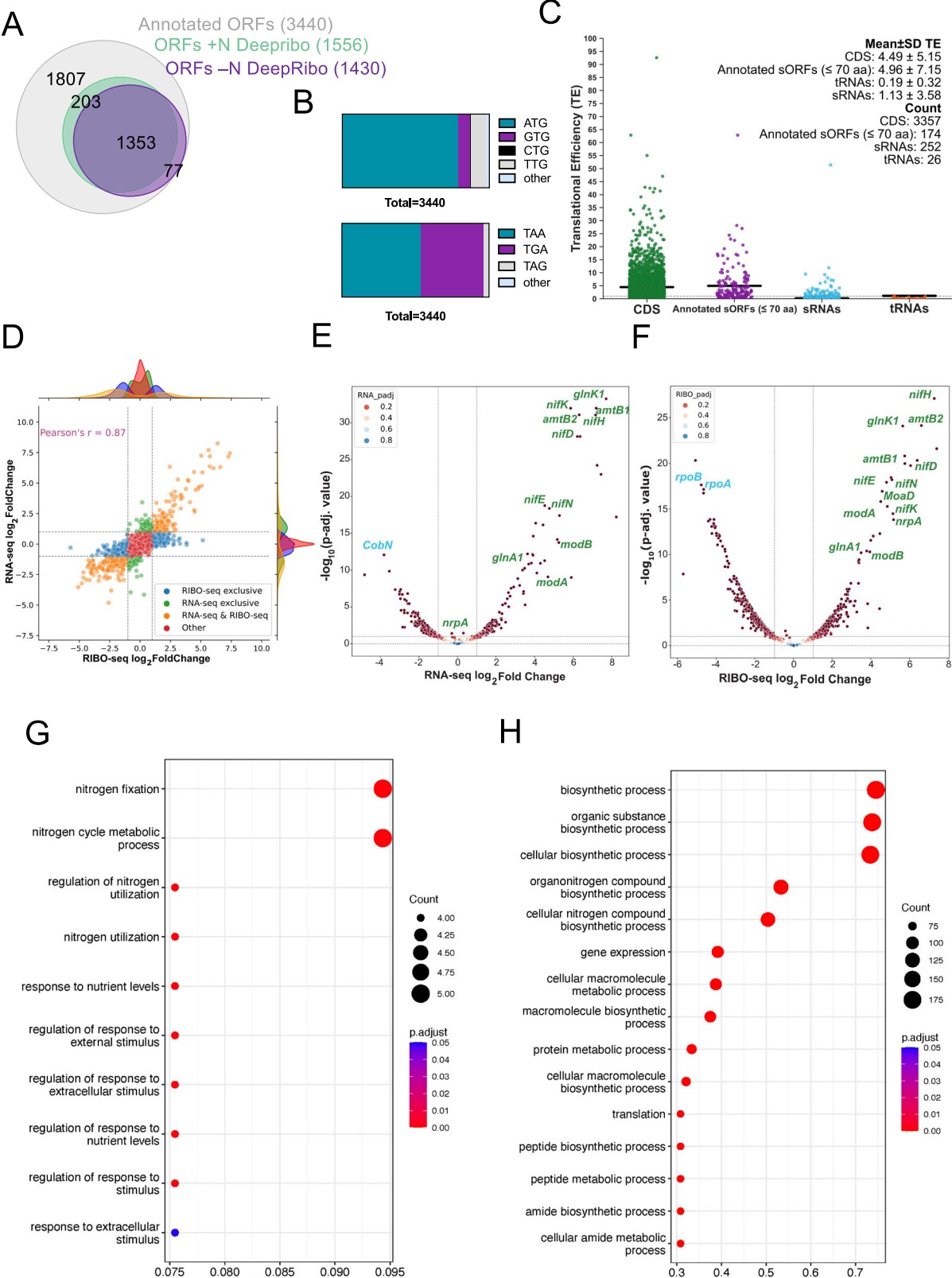

ferredoxin upregulated under N-limitation is very comprehensible, since nitrogenase reaction needs 16 electrons to reduce the molecular nitrogen ($N_2$) to ammonium, which are provided by reduced ferredoxins. Prediction tools for subcellular localization of the respective small proteins encoded by confirmed annotated sORFs indicate for 11%

to represent membrane proteins or to be located associated to the cytoplasmic membrane, whereas 73% are located in the cytoplasm. For the remaining 16%, no unique localization could be predicted (Fig. 4F). The translation of 62 annotated sORFs was further validated by MS (see Validation of unannotated candidates by LC-MS analysis).

**Fig. 3 | The global translatome. A** ORFs predicted to be translated by DeepRibo, a tool included in the HRIBO pipeline. To detect translation, we used the following parameters on the Ribo-seq data under +N and -N conditions: TE of ≥ 0.5 and RNA-seq and Ribo-seq RPKM of ≥ 10, when prediction by DeepRibo algorithm was needed, a positive DeepRibo score > 0 was chosen for cutoff criteria. **B** Start (upper panel) and stop codon (lower panel) distribution of annotated ORFs. **C** Scatter plot showing global translation efficiencies (TE = Ribo-seq/RNA-seq) computed from *M. mazei* Ribo-seq replicates under standard growth conditions, for all annotated coding sequences (CDS), annotated sORFs encoding proteins of ≤ 70 amino acids (aa), annotated small RNAs (sRNAs), and tRNAs. The black lines indicate the mean TE for each transcript class. Moreover, Mean ± SD TE and count of each transcript class is shown on the right top corner of plot. **D** Comparison of global RIBO and mRNA log2 FC values for +N and -N. Dashed lines indicate log2 fold change values of +1 or −1. Hundreds of genes exhibited differential expression (absolute log2 FC ≥ 1

and *p*-adjust ≤ 0.05) at both the transcriptional and translational levels (orange dots) with Pearson's correlation coefficient (r = 0.87), whereas others were exclusively detected by either RNA-Seq (green dots) or Ribo-Seq (blue dots). Volcano plots illustrating differential (**E**) mRNA levels of 185 downregulated genes and 152 upregulated genes and (**F**) RIBO levels 414 downregulated and 247 upregulated genes in *M. mazei* for all annotated ORF candidates identified in this study (Supplementary Data 2). The global translatome (**G**) Up- and (**H**) down-regulated biological processes in *M. mazei* under -N. Enrichment Analysis was conducted using the enrichGO function in the clusterProfiler package with the ribosome profiling differential expression data sorted by log2 fold change values as input. GO terms were considered as either up- or down-regulated if *p*-adjust values were ≤ 0.05. The top 15 non-redundant GO terms were sorted in descending order by the clusterProfiler gene ratio. Source data for **C** and **D** are provided as a Source Data file.

## Manual inspection of unannotated small proteome under +N and -N

The Ribo-seq based ORF prediction tool DeepRibo predicted approximately 23,500 sORFs (before the cut-off criteria was applied) encoding for small proteins ≤ 70 aa based on the two data sets obtained under + N and -N (Supplementary Data 4). We applied stringent cut-off criteria based on the analysis of the annotated sORFs (see Methods). This strict filtering led to 266 candidates, which were further investigated manually in the genome browser. As standard Ribo-seq dataset is not suited for detecting and predicting sORFs internal to longer genes, we decided to neglect the analysis of these predicted sORFs candidates. To detect those candidates a more sophisticated ribosome profiling method enriching the ribosomes at the translation initiation complex using antibiotics is required. After filtering and manual curation, we obtained a list of 63 unannotated small proteins. Additionally, to the manual inspection of these candidates, we observed the neighbourhood genes and searched for sORFs with a good Ribo-seq coverage. We found many sORFs which did not pass the stringent cut-offs, but following the criteria for manual inspection (see Methods), these unannotated sORFs were called as translated. After finalizing the analysis, we can provide a list of 314 translated small unannotated proteins with high confidence (Supplementary Data 5).

## Features of the unannotated sORFs

Overall, the majority of the 314 unannotated sORFs (Supplementary Data 5) were shorter than the annotated sORFs. Half of the respective unannotated small proteins have a length of 30 aa or less (Fig. 5A), whereas most of the annotated small proteins are more than 60 aa in length.

The start and stop codon distribution (Fig. 5B) shows that 56% of the sORFs start with the canonical ATG followed by TTG and GTG with approximately 20% each. Only 5% of the sORFs start with a CTG. The stop codon preference was in the order TAA (46%), TGA (36%) and at last TAG (18%). This distribution has changed towards non-canonical start and stop codons compared to annotated small proteins, which might be one reason why these sORF candidates have been overlooked in previous studies.

The 314 unannotated sORFs are encoded in different genomic context (Fig. 5C). Specifically, 34% of the unannotated sORFs are encoded in intergenic regions, 21% were found within previously detected sRNA candidates (Supplementary Data 5)[27], which indicates that the former reported sRNAs are either small mRNAs or dual-function RNAs with a regulatory RNA part and a part encoding a small protein. 10% of the unannotated sORFs were located in the 5'UTR of annotated genes, potentially representing uORFs in front of the respective long ORF and 9% in 3'UTR. 9% (Supplementary Data 5) were detected in the intergenic regions of operon structures and 15% in antisense orientation to annotated genes. 1% of the unannotated sORFs where either overlapping with an annotated gene or located within a

gene (in or out of frame) with a wrongly annotated start of translation, indicating for the need of further curation of the genome.

A differential expression analysis of 314 unannotated sORFs matched only 289 to the deltaTE output, revealing a significant upregulation of 20 and downregulation of 9 unannotated sORFs under -N at the RNA level, but 42 were upregulated and 7 were downregulated at the RIBO level under -N (Fig. 5D, E). The unannotated small proteins are predicted to be located in different regions of the cell (Fig. 5F). Around 40% of the small proteins are predicted to act in the cytoplasm of *M. mazei* whereas 3% of the small proteins are predicted to be associated to the cytoplasmic membrane or membrane proteins. Around 18% of the unannotated small proteins are predicted to act extracellular, however, no signal sequences for secretion could be predicted. A screenshot of sORF_16 and sORF_082 is shown as an example for differential expression in Fig. 5G. The translation of 26 unannotated sORFs was further validated by MS (see Targeted and untargeted in vivo validation of selected unannotated sORFs).

## Targeted and untargeted in vivo validation of selected unannotated sORFs

For validation by epitope tagging followed by western blot analysis, 16 small proteins were selected. The candidates were selected based on their different genomic context (Fig. 5C) or regulation in context of N availability and were tested under their respective native and/or a constitutive promoter. The two tested sORFs, sORF_03 and sORF_10, are located internal to annotated genes and are thus not included in the list of 314 unannotated sORFs.

Ten of the selected sORFs were episomally expressed in *M. mazei* under control of their respective native promotor and ribosome binding site (approximately 200 nt upstream of their predicted translation initiation site; see Methods) and fused to a C-terminal sequential affinity tag (SPA) for in vivo validation under their native transcriptional and translational regulation. The expression was tested under both nitrogen conditions. Out of those ten candidates, the translation of eight sORFs could be validated in a western blot analysis using a FLAG directed antibody against the SPA-tag (see Fig. 6A). Here sORF_03 showed a significant higher amount of expressed protein under +N conditions and in agreement with the Ribo-seq analysis sORF_09 was exclusively expressed under nitrogen sufficient conditions, whereas sORF_08 was only detectable under -N conditions, thus supporting the Ribo-seq analysis (see Fig. 5D, E). Corresponding growth analysis were performed for both nitrogen conditions for all ten selected strains together with an empty vector control (Supplementary Fig. 2A). A growth phenotype was detected for three *M. mazei* strains under -N condition (Supplementary Fig. 2B). The strains episomally expressing sORF_03 and sORF_15 under their native promoter showed a significantly longer lag phase and reached a lower overall growth density, just as the strain expressing sORF_08, which reached the stationary phase as well at a lower cell density than the control strain.

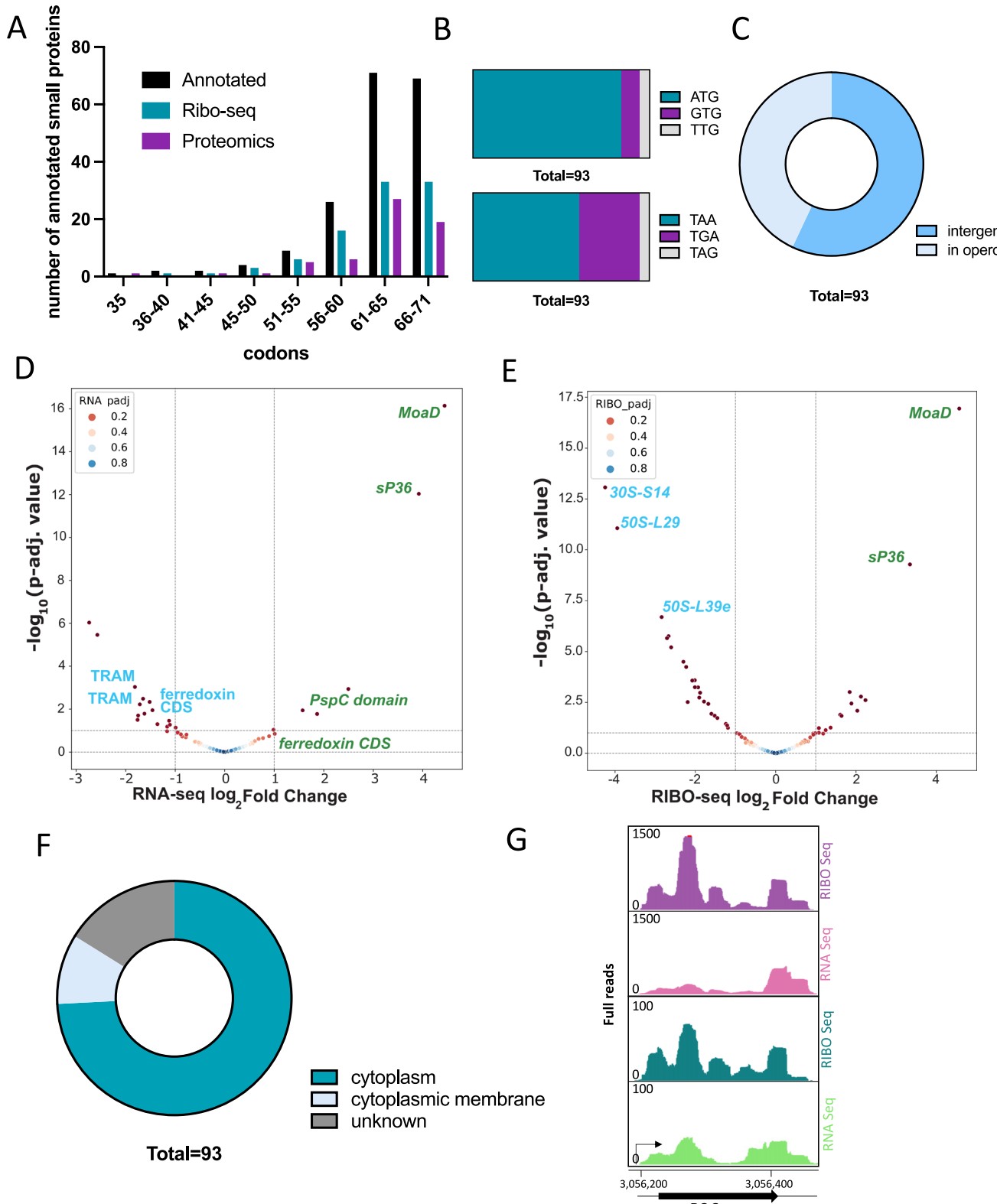

**Fig. 4 | Annotated small proteins in *M. mazei*. A** The histogram shows the codon length distribution of all annotated sORFs, the translated annotated sORFs and the aa of small proteins detected by mass spectrometry. **B** Start (upper panel) and stop codon (lower panel) distribution of annotated ORFs. **C** Genomic context of the translated annotated sORFs. Volcano plot for regulation of the small annotated sORFs at (**D**) transcription level of 11 downregulated and 5 upregulated genes and (**E**) translation level of 22 downregulated and 9 upregulated genes. **F** Subcellular localization of respective small proteins encoded by translated annotated sORFs. **G** sP36 as example for differential expression. Green color indicates +N condition, pink is -N condition (pay attention to different scales in y-axes).

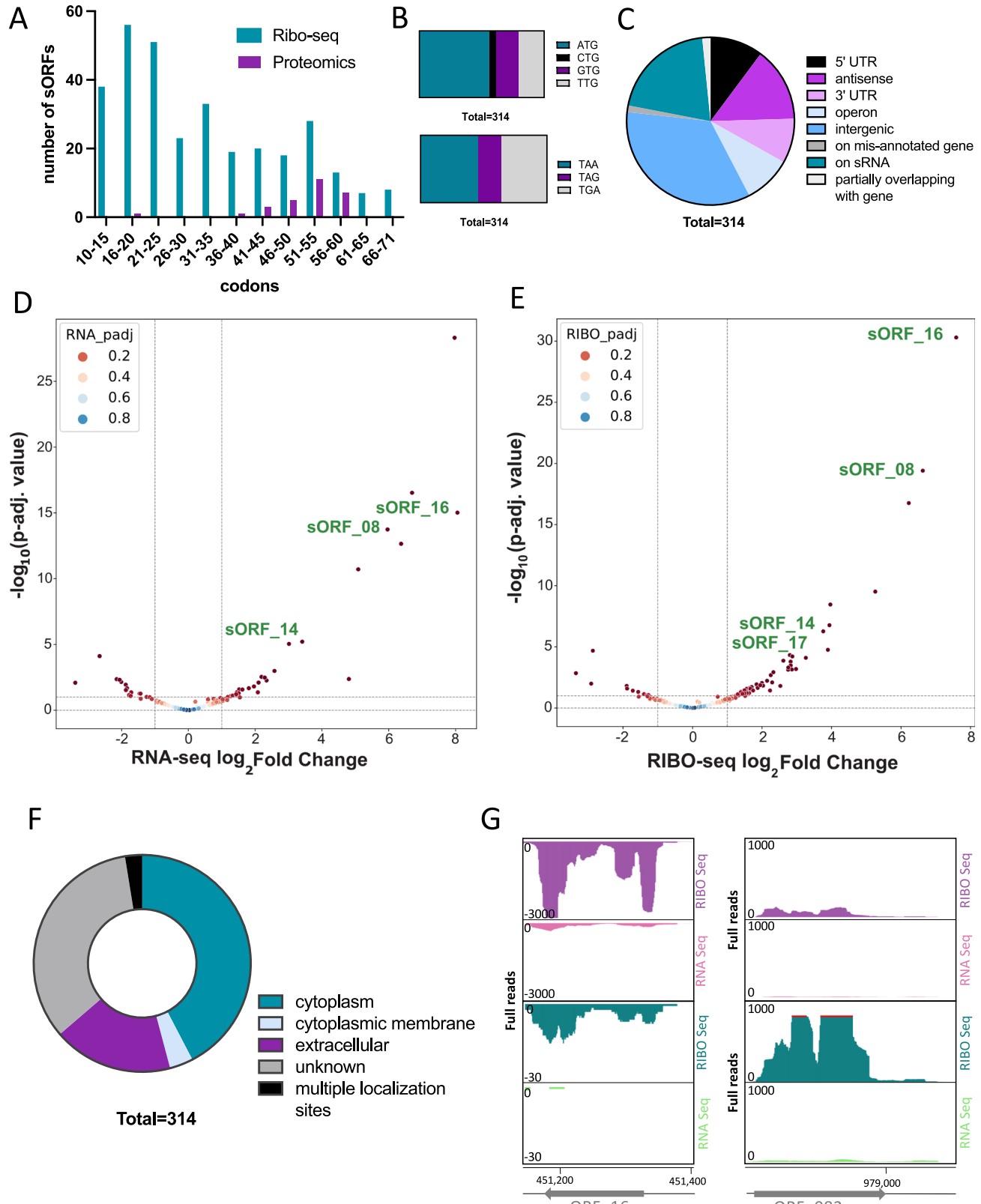

**Fig. 5 | Features of the unannotated small proteome. A** Length distribution of 314 sORFs based on Ribo-seq data and 26 small proteins identified by LC-MS. **B** Start (upper panel) and stop (lower panel) codon usage. **C** genomic context (**D**, **E**) differential expression (289 matched out of 314 unannotated sORFs). Volcano plot for regulation of the unannotated sORFs at (**D**) RNA level of 9 downregulated genes and 20 upregulated genes, and (**E**) RIBO level of 7 downregulated genes and 42 upregulated genes. **F** Localization in the cell (**G**) Screenshots of sORF_16 and sORF_082 as examples for N-regulation, pink/purple: -N, green/turquois: +N (pay attention to different scales in y-axes).

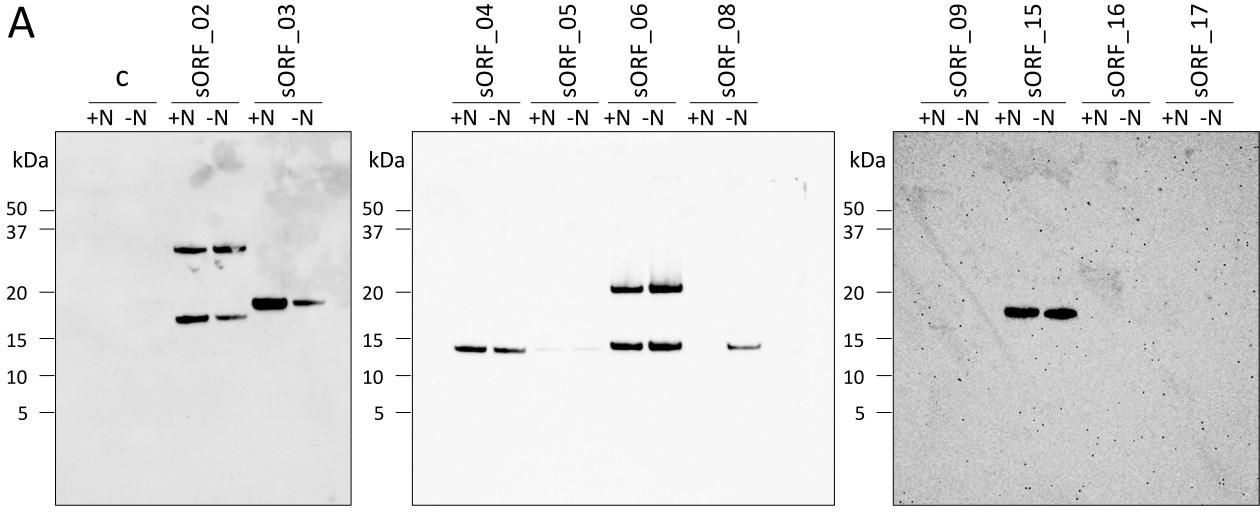

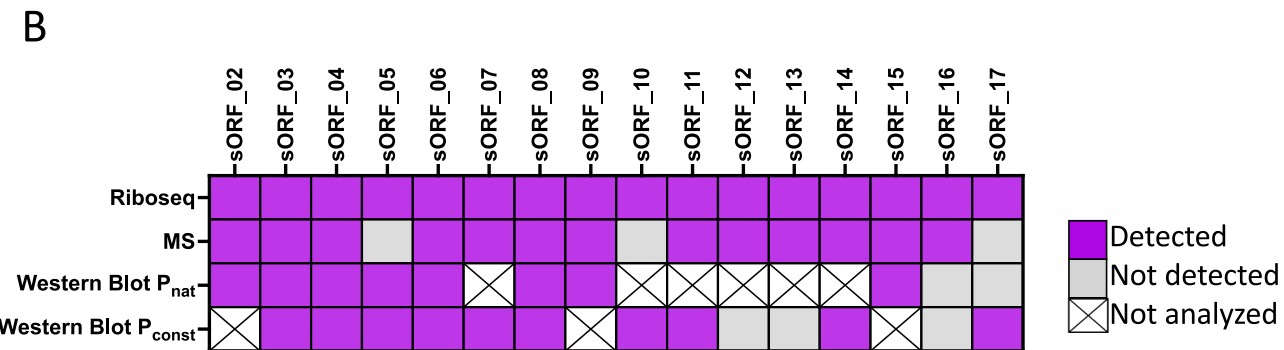

**Fig. 6 | Detection of small proteins in *M. mazei* cell extract by western blot analysis. A** Expression of selected small proteins under control of their respective native promoter and ribosome binding site under +N as well as –N conditions. The empty vector was used as a negative control. SDS PAGE followed by western blot analysis was performed with each 30 μg *M. mazei* cell extract from exponential growing cultures. A monoclonal FLAG-directed antibody was used to detect the SPA-tag at the C-terminus of the small protein under +N with constitutive overproduction. All corresponding growth experiments were performed in three biological replicates and the subsequent western blot was performed with the cell extracts from one biological replicate each. Source data are provided as a Source Data file. **B** Overview of detection using different methods for selected small proteins. Those which were not confirmed by western blot analysis have been validated by LC-MS.

Not all 16 selected sORFs could be cloned under their native promoter. In order to cover all selected 16 sORFs for in vivo expression analysis, another 13 out of those 16 sORFs were episomally expressed in *M. mazei* under the control of the constitutive P*mrcB* promoter and a standard *M. mazei* ribosome binding site. The expression was tested under nitrogen sufficiency conditions. Out of those 13 candidates, the translation of ten sORFs could be validated in a western blot analysis using a FLAG directed antibody against the SPA-tag (Supplementary Fig. 3 and 4 respective loading controls). In general, the detected molecular mass predicted upon the protein fragments detected in the western blot analyses correlates with the predicted aa based molecular mass of the respective protein together with the SPA-tag. In addition, sORF02 and sORF06 appear to be present as monomer and dimer.

An untargeted validation was performed via the re-analysis of previously generated proteomics datasets for *M. mazei*[54–57]. Reanalysis of these datasets was performed against an extended *M. mazei* protein database that included the unannotated sORF encoded small proteins. Reanalysis of the bottom-up proteomics dataset, which can provide a more sensitive analysis, allowed for the inferred identification of 62 previously annotated small proteins, and 26 encoded by unannotated sORFs. While further assessment of the previous top-down proteomics datasets allowed for the verification of 26 unannotated small proteins of which three were only identified via top-down analysis. Additionally, top-down proteomics analysis allowed for the confirmation of full length proteoforms with many of small proteins identified either with or without N-terminal excision of the first methionine residue. The identification of methionine excision on many of the small proteins further indicates that they undergo processing following translation. A detailed list of proteoforms confidently identified for the unannotated small proteins, as well as visualization of the fragmentation patterns for the most prominent protoforms, is provided (Supplementary Data 6 and Supplementary Fig. 5). Additionally, for those unannotated small proteins only identified via bottom-up analysis, or via top-down with less than 30% of residues cleavages mapped, a detailed list of the peptides identified via bottom-up analysis is provided (Supplementary Data 7).

In summary, validation of sORF expression under either the native or a constitutive promoter via epitope tagging and immunoblotting analysis was possible for 13 out of 16 selected sORFs. Nevertheless, the missing three sORFs (sORF12, sORF13, and sORF16) were detected by LC-MS (Fig. 6B).

## Phenotypic characterization of selected unannotated small proteins

The differential expression of sORF_03, sORF_08 and sORF_9 under the two N conditions gives a hint about a potential role in N regulation. To learn more about the potential function of selected unannotated small proteins, their conservation on the nucleotide level within the archaea was studied using tBlastn. Using the genes encoding the two small ribosomal proteins (MM_0340 and MM_1808) as comparison, this demonstrated sORF_10 and sORF_17 encoded small proteins are both exclusively found in *M. mazei*, the other small proteins are conserved (see Supplementary Fig. 6) in the genus of *Methanosarcina*. Moreover, homologues with lower confidence of small proteins encoded by sORF_03, sORF_04, sORF_05, sORF_06, sORF_12 and sORF_14 are also found in the family of *Methanosarcinacea*. The aa sequences of sORF_05 and sORF_06 encoded small proteins and their homologues based on tBlastn search were aligned by using ClustalOmega[58] depicted in Supplementary Fig. 7. The tBlastN search for sORF_05 showed that another copy is present in the genome, and that the homology applies not only applies to sORF_05, but also to the complete operon and the operon upstream of it in *M. mazei* strains indicating potential duplication of the operon region.

We further performed growth experiments with *M. mazei* episomally expressing selected sORFs under -N and +N. Most of those strains did not show any growth phenotype compared to the empty vector control (Supplementary Fig. 3). However, overexpressing sORF_05 had a negative impact on growth rate and lead to earlier transition into stationary phase in +N condition (Fig. 7A), whereas sORF_06 had a similar phenotype in −N condition (Fig. 7B). These results point to an important function of the respective small proteins in the cell and probably even in N metabolism. We thus concentrated on those two small proteins for further characterization.

For sORF_05 prediction methods as psortB and TMHMM gave hints for a potential membrane associated small protein. Moreover, structure predictions with AlphaFold 2[59,60] predicted a hydrophobic α-helical structure known for membrane integrated proteins (Fig. 7C). To investigate the membrane association of sORF_05 encoded small protein in vivo, cytoplasmic membranes of *M. mazei* overexpressing sORF_05 were separated from the cytoplasm and western blot analysis of the two fractions was performed using antibodies directed to the C-terminal SPA-tag. sORF_05 encoded small protein was mainly detectable in the membrane fraction and to smaller amounts (approx. 20 %) also in the cytoplasmic fraction strongly indicating a membrane association of the small protein (Fig. 7D). This is in accordance with very low amounts of detected protein in the cell extract after episomally expression under its native promoter (Fig. 6A). sORF_05 is located within a dimethylamine degradation operon together with a trimethylamine permease (MM_1691) (Fig. 7E), and a hypothetical protein (MM_1692); and has been added to the newest RefSeq annotation as MM_RS18880. Based on its genomic location it is attractive to speculate that the sORF_05 encoded small protein interacts or is an integral subunit of the trimethylamine permease, a membrane integrated transporter.

Based on prediction methods, the sORF_06 encoded small protein is localized in the cytoplasm of the cell. Structure prediction with AlphaFold 2 further showed the contiguity of the four conserved cysteines in the small proteins' amino acid sequence (Figs. 7F, S3). Clustered cysteines are known from proteins carrying an iron sulfur cluster as cofactor. sORF_06 is an intergenic unannotated sORF (standalone), thus we cannot learn much about its function from the genomic localization (Fig. 7G). Top-down proteomics allowed for unambiguous confirmation of sORF_06 translation, and, furthermore, that the respective small protein is translated in its entirety (i.e., the small protein was detected as a full length proteoform without subsequent truncation of the N or C termini), in addition it was identified with all four cysteine residues involved in disulphide linkages.

## Discussion

The intricate world of the small proteome ( < 71 aa), a largely uncharted component of the archaeal cellular apparatus, is brought to light in this study for the archaeon *M. mazei* Gö1. We present an integrated approach that combines Ribo-seq, computational predictions, filtering, and manual curation under two different growth conditions. This is complemented by LC-MS-based validation and additional in vivo protein validation, together providing an exhaustive portrayal of the small protein repertoire of *M. mazei*. Conducting comparative translatomics with Ribo-seq under two conditions ( + N and -N) in archaea is rare and has only recently been carried out in the first bacterium, *E. coli*, where the focus however was on the large proteome[61].

Ribo-seq has emerged as a potent method to study protein synthesis, and certainly boasts several advantages over proteomics[62,63]. However, it is not advisable to rely solely on Ribo-seq due to the inherent challenges, including complex ribosome behaviours, biases in library preparation, and differential ribosome occupancy[63,64]. Consequently, to enhance the reliability of Ribo-seq data, it's essential to combine it with manual curation and filtration of both Ribo-seq and RNA-seq datasets, as well as with validation methods like MS based proteomics and in vivo validation of proteins, as used in recent studies[39,40,45,63].

For bioinformatic analysis, the HRIBO workflow[48], was successfully used for the downstream analysis of Ribo-seq data. To identify robust unannotated sORF candidates we used DeepRibo prediction results instead of REPARATION, because DeepRibo predicted 37% more annotated sORFs than REPARATION (Fig. 3A). A benchmarking study by Clauwaert, et al.[49] found that DeepRibo surpassed REPARATION and other methods in terms of accuracy and sensitivity in performance across seven datasets. This more robust performance aligns with our choice of using DeepRibo for predictions. As reported by Gelhausen, et al.[65], DeepRibo is prone to a high rate of false positives, which we detect in our results as well, where DeepRibo predicted eight false positive out of 63 translated annotated sORFs and 57 false positives out of 255 predicted unannotated sORFs (Supplementary Data 4). Consequently, manual curation of the DeepRibo predictions with stringent cut-off filters based on control proteins (e.g. MM_RS08540) on prediction outputs was performed and provided more confident results in our study. The manual confirmation of translation status for the filtered DeepRibo predicted outputs for annotated ORFs, annotated sORFs and unannotated sORFs. One major challenge in the data processing is that the effectiveness of bioinformatic workflows like DeepRibo, REPARATION, deltaTE (for differential expression analysis), which are all optimized for bacterial genomes, might be limited on archaeal datasets. However, our optimized workflow using two conditions generated a dataset which provides an important resource for training these algorithms for archaeal data, suggesting a larger and more comprehensive *M. mazei* small proteome than currently understood.

One of the common sources of gene misannotation is the incorrect determination of translation initiation sites. This issue can be specifically addressed with Ribo-seq[66], since e.g. unconventional translation events are illuminated, enhancing the accuracy of gene annotations[67]. Ribo-seq revealed that 51 ORFs initiate translation downstream of the previously annotated start codons in *M. mazei*, suggesting N-terminal truncations in comparison to earlier annotations (Fig. 2B depicting one example). The remaining 61 ORFs out of 112 misannotated genes that were either divided into two smaller ORFs or showed a shift in their reading frames, which further emphasizes the role of Ribo-seq in accurately determining the coding potential of a genome. Moreover, the November 2022 Refseq annotation update of the *M. mazei* strain Gö1 genome validated our findings by incorporating corrections for 55% (61 out of 112) of the ORFs we identified as misannotated. This further reinforces the reliability and significance of our findings, emphasizing the value of our study in improving the accuracy of the genome annotation for *M. mazei* Gö1.

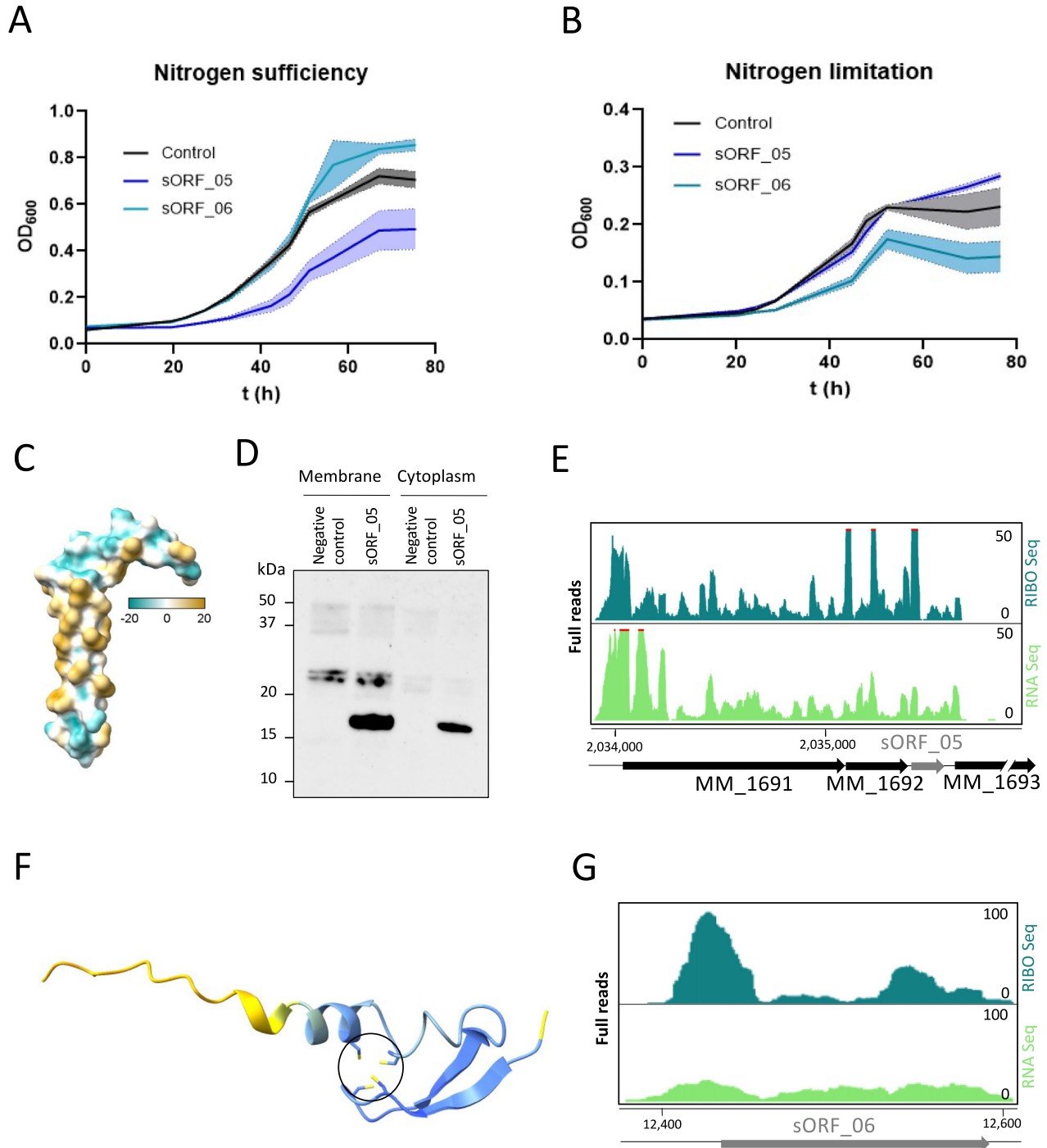

**Fig. 7 | Characterization of two unannotated small proteins encoded by sORF_05 and sORF_06. A** Overexpressed sORF_05 has a negative impact on *M. mazei* growth rate under N sufficiency, while (**B**) overexpressed sORF_06 leads to faster transition to stationary phase under N limitation. Growth of *M. mazei* mutants were monitored by measuring at $OD_{600}$ over the time. Data are represented as mean values ± standard deviation (SD) (visualized as shadows) is based on three biological replicates. Source data are provided as a Source Data file. **C** Structure prediction with AlphaFold 2 indicates a hydrophobic α-helical structure of sORF_05 encoded small protein. Colour shows hydrophobicity as indicated. **D** Membrane association of sORF_05 encoded small protein was validated via western blot of fractionated cell extract; one of two biological replicates is exemplarily shown. Source data are provided as a Source Data file. **E** Screenshot of the TMA permease transporter operon shows the location of sORF_05 integrated in the operon structure (Turquoise, Ribo Seq; green, RNA Seq. Scale bar shows full reads). **F** Predicted structure of sORF_06 encoded small protein by AlphaFold shows four clustered cysteins (highlighted in yellow, in black circle). Colour shows confidence (Blue, high; yellow, low). **G** Screenshot from JBrowse shows the high translation of sORF_06.

Ribo-seq effectively detects global translation with high sensitivity, yet unlike its performance in eukaryotic systems it falls short of achieving codon resolution in prokaryotes (bacteria or archaea). This represents a major challenge in achieving codon resolution in prokaryotic genomes[40,45,47,68,69]. Our optimized Ribo-seq workflow for *M. mazei* (see Methods), resulted in the accurate MNase restriction of 5' and 3' regions of translated mRNAs (Figs. 1C, D, S9). Our Ribo-seq libraries, obtained under both +N and -N conditions, exhibit a wide range of footprint lengths spanning from 15 to 40 nucleotides, with a prevalent occurrence of footprints measuring 23-25 nucleotides

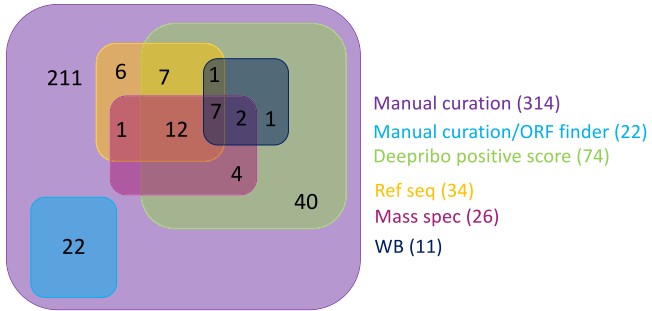

**Fig. 8 | Overview of detection methods and validation for 314 unannotated sORFs after manual curation.** 22 sORFs were predicted with NCBI ORF finder, 74 sORFs had a positive prediction score from Deepribo prediction, 34 sORFs are included in the actual Ref seq annotation (November 2022), 26 sORFs were validated by MS analysis and 13 out of 16 sORFs were confirmed by western blot analysis and two of them are not included in the unannotated sORFs list (internal to the ORFs).

**Table 1 | Validation of previously published results for genes involved in nitrogen metabolism**

| | Published[75] | This Study | |
|---|---|---|---|
| Genes | Transcription FC ± SD -N/ + N | Transcription FC ± SD -N/ + N | Translation FC ± SD -N/ + N |
| *nifH* | 197.1 ± 89.7 | 143.43 ± 3.62 | 153.69 ± 4 |
| *glnK₁* | 15.6 | 209.39 ± 3.71 | 49.62 ± 3.39 |
| *amtB₁* | 167.4 ± 37.5 | 146.34 ± 3.75 | 53.58 ± 3.74 |
| *glnA₁* | 26.5 ± 9.3 | 14.3 ± 3.53 | 11.18 ± 3.34 |
| *glnA₂* | 0.96 | 1.61 ± 3.53 | 0.62 ± 3.32 |
| *nrpA* | 39.6 ± 9.6 | 1.24 ± 4.39 | 35.7 ± 3.84 |
| *nifD* | 213.7 ± 107.2 | 74.39 ± 3.46 | 65.8 ± 3.96 |
| *nifK* | 202.5 ± 77.7 | 58.64 ± 3.24 | 34.92 3.95 |

(Supplementary Fig. 8). Higher TE values of annotated CDS in comparison to non-coding sRNAs and tRNAs (Fig. 3C) and pronounced ribosome protection of up to 16 nt upstream and downstream of start and stop codons (Fig. 1C, D) clearly demonstrated the successful establishment of Ribo-seq for *M. mazei*. Similar validation metrics have recently been reported in the successful establishment of Ribo-seq for *Sinorhizobium meliloti* in one standard growth condition[45].

LC-MS analysis confirmed the translation of 62 annotated sORFs predicted by Ribo-seq (Supplementary Data 6 and 7). While Ribo-seq, complemented by manual curation and gene neighborhood filtering, identified 314 unannotated sORFs, 26 of which were substantiated at the protein level by LC-MS, suggesting that relying exclusively on Ribo-seq or LC-MS approach likely underestimates the total number of sORFs, again highlighting the importance of in vivo validation to reinforce predictions from either method (Fig. 8). Furthermore, we studied in vivo expression of sORFs by epitope-tagging the respective sORFs under the control of their native promoter. Here we obtained eight out of ten tested sORFs, from which three showed strong nitrogen regulation (see Fig. 6A). Particularly the membrane-bound unannotated small proteins were not detected by LC-MS but validated via epitope tagging and immunoblotting analysis, e.g., sORF_05 (Fig. 6B), which illustrates the analytical strength of our combinatorial approach for detecting small proteins. Considering that current genomic annotation algorithms exhibit bias towards longer ORFs, consequently leading to the systematic underrepresentation of sORFs in genomic databases[70], this experimentally validated Ribo-seq dataset now allows for refining the small proteome of *M. mazei* as previously described in *E. coli* and *Staphylococcus aureus*[7,71,72]. Additionally, the *M. mazei* strain Gö1 Refseq annotation from November 2022 subsequently

incorporated 34 out of the 314 unannotated sORFs predicted in our study using the Genbank2014 annotation. This further emphasizes the high quality of our data (Supplementary Data 5).

In addition to providing comprehensive genome-wide archaeal Ribo-seq analysis for a strictly anaerobically growing archaeon it is important to point out that this is a ribosome-binding map under two growth conditions ( + N and -N). Our Ribo-seq analysis demonstrated that, across +N and -N conditions, 1556 and 1430 ORFs were actively translated, representing 45% and 42% of the 3440 annotated ORFs, respectively. Overall, 1633 ORFs showed translation in either or both conditions, with 1353 ORFs common to both, 77 unique to -N, and 203 exclusive to +N conditions (Fig. 3A). Moreover, the Ribo-seq analysis identified 93 and 95 translated annotated sORFs under +N and -N conditions, respectively, accounting for 51% and 52% of the total 184 sORFs coding for small proteins ( < 71 aa), with a combined total of 96 sORFs translated across conditions, 92 shared between both, three specific to -N, and one exclusive to +N condition (Supplementary Data 3). Although this study examines two conditions, the detection of 37% annotated sORFs in *M. mazei* aligns with previously observed ranges in *S. meliloti* (33%, Hadjeras, et al.[45]), and in *E. coli* (40%, Weaver, et al.[73], yet falls substantially below those found in *Haloferax volcanii* (65%, Hadjeras, et al.[40]), and *Salmonella* (76%, Venturini, et al.[69]) The discovery of 314 unannotated sORFs in *M. mazei* dramatically outnumbers those identified in other prokaryotes using exclusively Ribo-seq alone or combined with LC-MS, such as *Salmonella* (42, Venturini, et al.[69]), *E. coli*, (68, Weaver, et al.[73], *H. volcanii* (48, Hadjeras, et al.[40]), and *S. meliloti* (48, Hadjeras, et al.[45]) However, the number of previously annotated sORFs in *M. mazei* (184), of which 93 were confirmed by manual curation, is much smaller than in other organisms. Together, we can now provide a list of 407 small proteins with high confidence which represent approximately 12% of total ORFs. In bacterial genomes, a recent study showed that 16% ± 9% of total ORFs are sORFs[74], indicating that a similar ratio can be expected for archaea. Overall, the variation in the detection of unannotated sORFs across the different studies and reports is likely a product of unique experimental designs, computational methodologies, and inherent biological differences between organisms (including the genome size).

Genes involved in N metabolism in *M. mazei* have been shown to be transcriptionally regulated in response to N availability by a global repressor NrpR[23,24]. Those include the structural genes of nitrogenase (*nifHDK* genes), *glnA₁* encoding glutamine synthetase, the *glnK₁ amtB₁* operon encoding the nitrogen sensing P-II like protein (GlnK₁) and the ammonium transporter B₁, and the *nrpA* encoding the *nif* specific activator[75]. The reported differential expressions are mainly in agreement with the results of the current study (Fig. 3E, F) summarized and compared in Table 1. In addition to transcriptional regulation, we identified one regulatory RNA (sRNA₁₅₄) as a central post-transcriptional regulator in the nitrogen metabolism in *M. mazei*[26]. Under nitrogen limitation loop 2 of sRNA₁₅₄ activates translation of the *nrpA* mRNA, whereas in case of *glnA₂* mRNA loop2 is masking the ribosome binding site and thus inhibits translation initiation[26]. In agreement, our current study confirms high up-regulation of sRNA₁₅₄ under -N (Supplementary Figs. 9E and 10A) and demonstrates a notable decrease in the translational efficiency of *glnA₂* under -N (TE of 10.41) compared to +N (TE of 24.25) as shown in Supplementary Fig. 10B, and Table 1. Moreover, the effect of post-transcriptional regulation of *nrpA*-mRNA by sRNA₁₅₄, enhancing translation efficiency as proposed by Prasse, et al.[26] is as well confirmed by the current data set (see Table 1 and Fig. 3E, F). Of particular note is that sRNA₁₅₄ is one of the sRNAs where the current analysis identified a leaderless sORF (Supplementary Fig. 11). This sORF_154 encodes a small protein of 20 aa (nct 4-66), which is highly conserved in *Methanosarcina* (Supplementary Fig. 6). Consequently, we propose that sRNA₁₅₄ represents a dual-function sRNA, which will be validated in the future.

Investigating the start codon distribution of annotated and unannotated sORFs, we observed that the usage of the canonical start codon ATG is reduced from 83% (annotated sORFs) to 56% (unannotated sORFs). In addition, annotated sORFs (with 60 aa or more in length) are in general longer than the unannotated sORFs, of which more than 50% are 40 aa or shorter. The identification of the very small sORFs and the sORFs with non-canonical start codons proves the importance of Ribo-seq, because other methods for ORF identification like proteomics and many pipelines for ORF detection are optimized for longer ORFs and have difficulties to identify those small sORFs (reviewed in ref. 76). Nevertheless, whether those very small proteins have a cellular function has yet to be shown.

The genomic context of sORFs might give a hint to their putative function. sORFs located in 5′UTR of longer genes often act as so called uORFs, influencing the translation of the downstream longer ORF[73] (see Supplementary Data 8). Other hints to a function can be the cellular localization of the encoded small protein. Membrane associated small proteins can act as toxins by forming pores in the membrane or interact with protein complexes in the membrane (reviewed in ref. 76). We found 64 out of 314 (21%) of the unannotated sORFs on previously published sRNAs (Table S5 from[27]), out of which 31 were already identified as so-called spRNA (small protein encoding sRNA), and others (33) were classified as regulatory sRNA. These translated sRNAs (see Supplementary Data 9) might be either unidentified dual-functional sRNA, regulating cellular processes based on a non-coding regulatory RNA part and an encoded small protein[41,43,77,5,78], or they are in fact spRNA (small mRNAs) and the putative sORF was overlooked by in silico prediction tools or manual inspection in previous studies. In this respect, Ribo-seq is a powerful tool to uncover these sORFs as it can show whether a sRNA is in fact a non-translated sRNA or a small mRNA. However, the Ribo-seq method faces limitations in identifying alternative sORFs internal to the longer ORFs, necessitating TIS-profiling, which has to be performed to stall ribosomes at initiation sites for accurate detection and correction of misannotated genes[47,73].

In conclusion, this study reveals a more extensive and dynamic small proteome than previously appreciated, underscoring the nuanced regulatory mechanisms at play in the cellular machinery of *M. mazei*. Our approach exemplifies the strength of combining multi-omic strategies, including Ribo-seq, to correct and enhance genome annotations, particularly for sORFs, which are often overlooked e.g., due to biases towards longer ORFs. This dual-condition translational landscape illuminates the underexplored small proteome of *M. mazei* Gö1 and demonstrates its plasticity across N-rich and N-depleted conditions, contributing to a refined understanding of archaeal proteomics. Our findings underscore the effectiveness of multi-omic analyses in overcoming biases against small ORFs, enhancing the precision of genomic annotations and providing a critical resource for future research into archaeal biology. By providing a more accurate representation of small proteins, which constitute around 12% of total ORFs in *M. mazei*, our study sets a latest standard for genomic annotation in archaeal organisms and offers a valuable resource for further biological and evolutionary studies.

## Methods

### Strains and plasmids
Strains and plasmids used in this study are listed in Supplementary Table 1 and 2. Plasmids were transformed into *E. coli* DH5α[79] or DH5α λpir[80] as previously described[81]. Briefly, the plasmid was added to competent *E.coli* DH5α cells. The cells were then incubated in an ice bath for 30 min, heat-pulsed at 42 °C for 30 s and then transferred to an ice bath. After 0.8 ml of SOC media was added, the cells were incubated at 37 °C with shaking for 1 h followed by plating on LB plates containing antibiotics. Transformation into *M. mazei*[29] was performed by liposome mediated transformation with modifications described in refs. 30,82. Briefly, inside an anaerobic chamber 30 μL of DOTAP were mixed with 70 μL of sucrose buffer (10 mM MES, 0.15 M sucrose, 6.3 pH) to create a DOTAP–sucrose mixture. 1 μg of DNA were diluted in 50 μL of sucrose and was then added to the 100 μL DOTAP–sucrose mixture and was let sit for 30 to 40 minutes. 10 ml *M. mazei* preculture was pelleted. The cells were resuspended in 980 μL of sucrose and then the 150 μL DNA-liposome complexes were added and incubated for four to six hours at 37 °C and supplemented with 5 mM MgCl₂ and 5 mM MnCl2 after 3 h. 500 μL of transformed cells were transferred to 0.1 M sucrose-containing minimal medium and incubated for 12 to 16 hours at 37 °C. After that, 200 to 300 μL of the culture was transferred into fresh sucrose and puromycin-containing medium for further growth and selection. Primers used in this study are listed in Supplementary Table 3.

### Cell growth and harvest for Ribo-seq
*M. mazei* Gö1 was cultivated either under +N or -N condition as described by Ehlers, et al.[20]. Briefly, *Methanosarcina mazei* Gö1 was grown without shaking at 37 °C, in 5- or 25-ml closed growth tubes using the Hungate technique on 150 mM methanol in minimal medium DSMZ 120 (methanosarcina media) (pH 6.9) with the following changes: K₂HPO₄ (Carl Roth, Germany), 0.456 g and cysteine hydrochloride (GERBU Biotechnik, Germany) 0.45 g each per litre. Growth generally took place in a nitrogen atmosphere containing 20% CO₂. For nitrogen-limiting growth conditions, ammonium was omitted from the medium. Control growth experiments for nitrogen fixation were performed in either an argon atmosphere or a hydrogen atmosphere containing 20% CO₂. Cells corresponding to 60 OD₆₀₀ equivalent units were harvested rapidly by fast-chilling in an ice bath to halt cell growth and translation, without the use of antibiotics, as described previously[40,45,69]. Briefly, cultures in the exponential phase (OD₆₀₀ 0.5) were rapidly placed in the ice-water bath and incubated with gentle shaking for 3 min. A culture aliquot was withdrawn for total RNA analysis, mixed with 0.2 vol stop mix (5% buffer-saturated phenol (Carl Roth, Germany) in 95% ethanol), and snap frozen in liquid nitrogen. The rest of the cultures were then immediately pelleted by centrifugation (10 min at 2455 × g at 4 °C). The obtained cell pellets were resuspended in 1 mL ice-cold media and centrifuged again at 16,100 x g at 4 °C for 10 min before snap-freezing in liquid nitrogen. We choose to harvest *M. mazei* cultures with no translation elongation inhibitors as such chemicals can introduce bias into Ribo-seq coverage[68,83]. The fast-chilling harvest method recovered stable polysomes.

### Preparation of *M. mazei* ribosome footprints
Ribosome profiling was performed as previously described[44] with some modifications. Briefly, frozen cell pellets were resuspended with cold lysis buffer (100 mM NH₄Cl, 10 mM MgCl₂, 20 mM Tris-HCl, pH 8.0, 0.4% Triton X-100, 0.1% NP-40, 150 U DNase I (Fermentas, Leon-Rot, Germany), 500 U RNase Inhibitor (MoloX, Berlin, Germany)) and lysed by sonication (constant power 50%, duty cycle 50%, 3 × 30 s cycles with 30 s cooling on a water-ice bath between each sonication cycle to avoid heating of the sample and RNA degradation). The lysate was clarified by centrifugation at 10,000 × g for 13 min at 4 °C. Next, 14 A₂₆₀ of lysate was digested with 20,000 U of micrococcal nuclease (MNase, NEB, USA) in lysis buffer supplemented with 10 mM CaCl₂ and 500 U RNase Inhibitor. Polysomes digestion was performed at 25 °C with shaking at 450 rpm for 60 min. A mock-digested control (no enzyme added) was performed in parallel to confirm the presence of polysomes in the lysate. MNase digestion was stopped with ethylene glycol-bis(β-aminoethyl ether)-N,N,N′,N′-tetraacetic acid (EGTA, final concentration 6 mM). To analyze polysome profiles and recover digested monosomes, equivalent A₂₆₀ units of lysates were layered onto a linear 10-55% sucrose gradient prepared in the following buffer: 10 mM MgCl₂, 20 mM Tris-HCl, pH 8, 100 mM NH₄Cl, 5 mM CaCl₂, 2 mM dithiothreitol (DTT), in an ultracentrifuge tube (13.2 mL Beckman Coulter SW-41). Gradients were centrifuged in a SW40-Ti rotor at

155,000 RCF (x g) for 2 h 30 min at 4 °C in a Beckman Coulter Optima XPN-80 ultracentrifuge. Gradients were processed using a Gradient Station (IP, Biocomp Instruments) fractionation system with continuous absorbance monitoring at 254 nm to resolve ribosomal subunit peaks. The 70S monosome fractions were collected and subjected to RNA extraction to purify the RNA footprints. RNA were extracted from the collected sucrose gradient fractions using hot phenol-chloroform-isoamyl alcohol (PCI) protocol[69,84]. Briefly, after thawing, 1% SDS was added to each sucrose fractions and incubated for 2 minutes at 64 °C. 1 volume of pre-warmed PCI (25:24:1, Carl Roth, Germany) was added to the sucrose fractions and incubated for 5 minutes at 64 °C without shacking. After 5 minutes incubation on ice, the samples were centrifuged 5 minutes at 13000 rpm at 4 °C. The aqueous phase containing RNA was transferred to a new tube and a second PCI extraction was carried out as described above but at 21 °C. After centrifugation for 5 minutes at 13000 rpm at 4 °C, the final aqueous phase was transferred to new tubes and RNA were precipitated by addition of 2 volumes of isopropanol, 1/10 volume of NaOAc (3 M, pH 5.5), and 1 µl of Glycoblue followed by incubation at −20 °C overnight. Total RNA were extracted from the cell cultures that were mixed with 0.2 volumes of stop solution (ethanol:phenol 95:5 v/v). Briefly, samples were thawed on ice, spun at 4000 rpm for 10 minutes at 4 °C & the resulting cell pellet was dissolved in 1 ml TRIzol (Invitrogen, #15596026). The mixture was transferred to 2 ml PLG tubes (Phase Lock Gel tubes-heavy (Eppendorf, #955154045) containing 400 µl chloroform. The samples were mixed by shacking and centrifuged at 14000 rpm for 15 minutes at room temperature. The aqueous phase was transferred to a new tube and nucleic acids were precipitated by addition of 1 volume of isopropanol followed by incubation at −20 °C overnight. After RNA precipitation and centrifugation at 14000 rpm for 30 minutes at 4 °C, the RNA pellets were washed with 75% ethanol, and centrifuged again at 14000 rpm for 15 minutes, The air-dried pellet was resuspended in milliQ water (RNase-free) and heated for 5 minutes at 65 °C. The total RNA preparation (40 µg) was treated with DNASe in order to remove contaminating DNA using DNase I (Invitrogen, #18068015), following the manufacturer's instructions. Ribosomal RNA was depleted from 5 µg of DNase I-digested total RNA by subtractive hybridization with the Pan-Archaea riboPOOLs (siTOOLs, Germany) according to the manufacturer's protocol with Dynabeads (MyOne Streptavidin T1 beads, Invitrogen, Germany). Total RNA was fragmented with RNA Fragmentation Reagent (Ambion), following the manufacturer's instructions. Monosome RNA extracted from sucrose fractions and fragmented total RNA were both denatured for 90 seconds at 80 °C in RNA loading dye (95% formamide, 0,1% xylene cyanol, 0,1% bromophenol blue, 10 mM EDTA), chilled on ice for 2 minutes, then separated on a 15% denaturing PAA gels (7 M urea) along with an ultra low DNA range ladder (Thermo Scientific) and a mixture of an upper and lower size marker (RNA oligonucleotides NI-NI-19 and NI-NI-20) as guides on either side of the RNA samples[85]. After electrophoresis for 70 minutes at 200 V, the gel was stained for 3 minutes with 1X SYBR Gold (Thermo Fisher Scientific) in 1X TBE running buffer. After gel visualization, the 20nt-34nt regions of the footprint RNA & total RNA demarcated by the oligo guides NI-NI-19 and NI-NI-20 & DNA ladder was excised in a clean 0.5 ml non-stick RNAse-free tube. The bottom of the tubes were pierced by a gauge needle and tubes were placed in a 1.5 ml tube and spun 2 minutes at full speed in a tabletop microcentrifuge to force the gel slice through the holes. 400ul RNA gel extraction buffer (300 mM NaAc pH5.5, 1 mM EDTA, 0.25% wt/vol SDS) were added to the tubes and incubated overnight at 8 °C with gentle mixing (650 rpm). After incubation for 15 minutes at room temerature, all liquid and gel debris were transferred to a SpinX centrifuge tube filter (VWR, cat. no. 29442-752). After centrifugation at 10000 rpm for 5 minutes at room temperature, RNAs were precipitated with 2 volume of isopropanol and 15 µg GlycoBlue (Ambion) overnight, cleaned up, and eluted with milliQ water (RNase-free) and then stored a −20 °C.

cDNA libraries were prepared by Vertis Biotechnologie AG (Freising, Germany) using the adapter ligation protocol without fragmentation. At first, an oligonucleotide adapter was ligated to the 3′ end of the RNA molecules. First-strand cDNA synthesis was performed using M-MLV reverse transcriptase (Moloney Murine Leukemia Virus Reverse Transcriptase; Promega) and the 3′ adapter as primer. The first strand cDNA was purified, and the 5′ Illumina TruSeq sequencing adapter was ligated to the 3′ end of the antisense cDNA. The resulting cDNA was PCR-amplified to about 10-20 ng/µl using a high-fidelity DNA polymerase. The DNA was purified using the Agencourt AMPure XP kit (Beckman Coulter Genomics) and was analyzed by capillary electrophoresis. The primers used for PCR amplification were designed for TruSeq sequencing according to the instructions of Illumina. The following adapter sequences flank the cDNA inserts: TruSeq_Sense_primer: (NNNNNNNN= i5 Barcode for multiplexing) 5′-AATGATACGGCGACCACCGAGATCTA-CAC-NNNNNNNN-ACACTCTTTCCCTACA CGACGCTCTTCCGATCT-3′; TruSeq_Antisense_primer: (NNNNNNNN= i7 Barcode for multiplexing) 5′-CAAGCAGAAGACGGCATACGAGAT-NNNNNNNN-GTGACTGGAGTT-CAGACGTGT GCTCTTCCGATCT-3′. cDNA libraries were pooled on an Illumina NextSeq 500 high-output flow cell and sequenced in single-end mode (75 cycles; with 20 million reads per library) at the Core Unit SysMed at the University of Würzburg, Würzburg, Germany.

## Ribosome profiling data analysis

*M. mazei* Ribo-seq data were processed and analyzed using the HRIBO pipeline (version 1.7.0)[48], which has previously been used for bacterial[45,69], and for archaeal Ribo-seq data[40]. Briefly, sequencing read files were processed with a snakemake[86] workflow that downloads all required tools from bioconda[87] and automatically determines the necessary processing steps. Adapters were trimmed from the reads with cutadapt (version 2.1)[88] and then mapped against the *M. mazei* genome with segemehl (version 0.3.4)[89]. Reads corresponding to ribosomal RNAs (rRNAs), transfer RNAs (tRNAs) and multiply mapping reads were removed with SAMtools (version 1.9)[90]. Quality control was performed by creating read count statistics for each processing step and RNA-class with Subread featureCounts (1.6.3)[91]. All processing steps were analyzed with FastQC (version 0.11.8)[92] and results were aggregated with MultiQC (version 1.7)[93].

Read coverage files were generated with HRIBO using different full read mapping approaches (global or centred) and single-nucleotide mapping strategies (5′ or 3′ end). Read coverage files using two different normalization methods were created (mil and min). For the mil normalization, read counts were normalized by the total number of mapped reads within the sample and scaled by a per-million factor. For the min normalization, the read counts were normalized by the total number of mapped reads within the sample and scaled by the minimum number of mapped reads among all the analyzed samples. The coverage files generated using the min normalization and the global mapping (full read) approach were used for genome browser visualization. To ensure the accuracy of the predictions, a rigorous filtering process was performed and manual curation of the predicted ORFs was carried out by comparing the cDNA read coverage of the Ribo-seq library with the expression signals obtained from the paired RNA-seq library, we were able to delineate features such as coding potential, ORF boundaries, and 5′- and 3′-UTRs.

## ORF prediction, filtration, and manual curation

Open reading frames (ORF) based on Ribo-seq were called with an adapted variant of REPARATION[50] using blast instead of USEARCH (see https://github.com/RickGelhausen/REPARATION_blast) and with DeepRibo[49]. Additionally, GFF track files with the same information were created for in-depth manual genome browser inspection, in addition to GFF files showing potential start and stop codon, and RBS information. Summary statistics for all available Genbank annotated and merged unannotated ORFs detected by REPARATION and

DeepRibo were computed in a tabularized form including amongst other values translation efficiency (TE), RPKM normalized read-counts, codon counts, nucleotide and amino acid sequences (see Supplementary Data 4).

Annotated ORFs were classified as translated if they fulfilled a TE cut-off ≥ 0.5 and RNA-seq and Ribo-seq RPKM ≥ 10 (cut-offs chosen based on the lowest TE and RPKM values associated with housekeeping genes (i.e., ribosomal protein genes) and the genes detected by proteomics). All the annotated sORFs were manually inspected to infer their translation and to assess the prediction tools power and to generate a positive set. To identify strong candidates for unannotated sORFs, we inspected HRIBO ORF predictions from DeepRibo and REPARATION. As DeepRibo is prone to a high rate of false positives[65], we first generated a reasonable set of potential unannotated sORFs by applying the following expression cut-off filters: meanTE ≥ 0.24 and RNA-seq and Ribo-seq RPKM ≥ 10 (in both replicates) based on the 93 positively labelled as translated annotated sORFs. In addition, unannotated translated sORF candidates were required to be predicted by DeepRibo with a prediction score > 0 that allows for ORF candidate ranking[49].

## Manual curation of sORF candidates

The filtered sORFs were then subjected to manual curation as previously described[45,65]. Briefly, inspection of the Ribo-seq coverage in a genome browser was conducted using these following criteria: **(i)** the shape of the Ribo-seq coverage over the ORF: the evenness of the Ribo-seq coverage was considered and any predicted sORFs exhibiting uneven coverage with peaks in the shape of a plateau, resulting from RNA structures and/or technical biases, we did not consider as an unannotated sORF; **(ii)** restriction of Ribo-seq coverage within ORF boundaries (and ribosome footprints excluded from 5′/3′UTRs) was required; **(iii)** the Ribo-seq signal was generally required to be comparable or higher to the transcriptome signal from the RNA-seq library and **(iv)** RNA-seq and Ribo-seq coverage was required to be, generally, at least ten reads per nucleotide normalized (RPKM) by sample size. Where DeepRibo was not able to predict a suitable sORF, we used NCBI ORFfinder (https://www.ncbi.nlm.nih.gov/orffinder/) to find the right sORF. In addition, during the manual inspection, some sORFs were found to be translated based on their coverage but did not pass our stringent filtering criteria. Those were added to the list of potentially translated sORFs. After finalizing the analysis, we could predict translation for 314 unannotated sORFs (The Ribo-seq datasets described in this study can be visualized in this JBrowse (http://www.bioinf.uni-freiburg.de/ribobase)[94]. Ribo-seq and RNA-seq data have been deposited in GEO with the accession number GSE240615.

## Validation of unannotated candidates by LC-MS analysis

Raw data files from previously published proteomics datasets[54–57] were subjected to reanalysis using the Proteome Discoverer software suite. In addition, a further *Methanosarcina mazei* dataset was generated following the protocols as per[54] with bottom-up analysis performed as per the procedure outlined in that publication. Briefly, 50 mg of wet weight *M. mazei* was lysed via repeated freeze/thaw cycles in 300 μl of 6 M guanidinium hydrochloride containing 1x complete protease inhibitor. The lysed material was heated to 72 °C for 15 minutes and then acidified with 3.7 ml 5% formic acid. The samples were then centrifuged at 21k × *g* for 20 minutes. The supernatant was pushed through a preconditioned SPE cartridge (3cc SepPak C18 200 mg). The SPE cartridge was washed (6 ml of 5% formic acid), and then the lower molecular weight fraction was eluted with 70% acetonitrile. The samples were split (1:9, BU:TD) and dried down via vacuum centrifugation in preparation for bottom-up or top-down proteomic analysis. For bottom-up proteomic analysis, approximately 20 μg of dried proteins were reduced with dithiothreitol (10 mM, 56 °C, 1 hr), alkylated with chloroacetamide (50 mM, 20 °C, 30 min), and then digested with trypsin (enzyme to protein ratio of 1:50, in 100 mM triethylammonium

bicarbonate buffer pH 8.0, overnight at 37 °C). The peptides were further cleaned up via SPE (1cc SepPak C18 50 mg), dried down via vacuum centrifugation and resuspended in loading eluent (3% acetonitrile, 0.1% trifluoroacetic acid) immediately before LCMS analysis. Bottom-up peptide separation was performed on a Dionex U3000 nanoHPLC system fitted with an Acclaim PepMap100 C18 column (75 μm × 500 mm). A C4 precolumn (PepMap100, C18, 5 μm) was used for sample loading. The eluents used were; eluent A: 0.05% formic acid in MilliQ water, eluent B: 80% acetonitrile, 0.05% formic acid, in MilliQ water. The separation was performed over a programmed 215-minute run. Initial chromatographic conditions were 4% B for 3 minutes followed by a linear gradient from 4-40% B over 180 minutes, 40-90% over 7 minutes, and 9 minutes at 90% B. Following this, an inter-run equilibration of the column was performed. Data acquisition following separation was performed on a Q Exactive Plus mass spectrometer. A full scan MS acquisition was performed (350-1600 *m/z*, resolution 70,000) with subsequent data-dependent MS/MS of the top 10 most intense ions via HCD activation at NCE 27 (resolution 17,500), dynamic exclusion was enabled (60 s). In addition, the samples were also subjected to top-down analysis following the optimized protocols established by Kaulich, et al.[95]. Briefly, for top-down LC-FAIMS-MS analysis, the dried proteins were resuspended in loading eluent immediately before analysis. Separation of the proteins was performed on a Dionex U3000 UHPLC system fitted with a C4 analytical column (50 cm × 75 μm, 2.6 μm, 150 Å). A C4 precolumn (C4 PepMap300, 5 μm, 300 Å) was used for sample loading. The eluents used were; eluent A: 0.05% formic acid in MilliQ water, eluent B: 80% acetonitrile, 0.05% formic acid, in MilliQ water. Separation was performed over a programmed 150-minute run. Initial chromatographic conditions were 4% B for 5 minutes followed by a linear gradient from 4-15% B over 2 minutes, 15-60% B over 120 minutes, 60-90% B over 2 minutes, and 11 minutes at 90% B. Following this, an inter-run equilibration of the column was performed. Data acquisition following separation was performed on a Fusion Lumos Tribrid mass spectrometer with a FAIMS Pro interface attached. A multi-CV methodology was used for data acquisition[95]. Source fragmentation was enabled (15 V), and Four FAIMS CV settings (−60, −50, −40, −25) were used in a data-dependent method (3 s cycle time), charge states 4-50 were included, and dynamic exclusion (60 s) was enabled. For CVs −60 and −50, the following settings were used: MS1 resolution 60 K, IT 118 ms, 2 microscans, MS2 resolution 50 K, max IT 200 ms, 2 microscans AGC target 4e5. For CVs −40 and −25: MS1 resolution 120 K, IT 246 ms, 4 microscans. MS2 resolution 60 K, max IT 250 ms, 4 microscans. Ion activation was performed with CID (25%). Datasets have been divided based on sample processing performed prior to LC-MS analysis. No specific enrichment/depletion pre-fractionation steps were performed for "Full_proteome" datasets, while datasets for the "Depletion_methods" searches derive from sample preparation methods in which enrichment/depletion pre-fractionation steps were performed.

Raw data files were searched against the protein FASTA file provided containing the full UniProt canonical proteins plus isoforms (accessed from UniProt:2023.03.24) and a curated list of predicted small proteins. This list contains all 184 annotated small proteins, and 314 unannotated small proteins encoded by the unannotated sORFs identified in Ribo-seq prediction and manual curation.

Bottom-up proteomics datasets were interrogated using Proteome Discoverer (Ver. 3.0.0.757). Three processing search nodes were employed to allow comprehensive data analysis. For the Chimerys node, a full tryptic search was employed with a fragment tolerance of 20 ppm, while for SequestHT with Inferys rescoring (full tryptic), and SequestHT (semi-tryptic), precursor and fragment tolerances of 10 ppm and 0.02 Da were employed. For all searches, variable oxidation of methionine was allowed, and fixed cysteine carbamidomethylation was set. For the SequestHT semi-tryptic search variable N-term formylation was additionally allowed. Data were merged using the

percolator node and the search files generated from data processing were combined and processed through consensus workflows resulting in the generation of three main results files; results from full proteome analyses (MmFull_Proteome_Methods.pdResults), results from depletion methods analyses (MmDepletion_Methods.pdResult). For the individual searches a strict peptide and PSM false discovery rate (FDR) was applied (< 1%). At the protein level, a less stringent false discovery rate was used for the reporting of protein groups (FDR < 5%). To ensure stringent data quality, the peptide spectral matches (PSM) of peptides mapping to small proteins were manually inspected. In cases where only a single (tryptic) peptide could be identified for a given small protein, an ion series of five consecutive b- or y- ions was required[96]. The FDRs reported are for individual searches, with the full lists comprised of proteins that met these criteria in the local searches, as opposed to a global FDR. An additional search using only Chimerys and SeQuestHT (semi-tryptic), and results from depletion searches under either nitrogen-limited or nitrogen-sufficient growth was also performed to evaluate potential differences arising from the two conditions (Mm_Depletion_N_limit_VsN.pdResult).

Top-down proteomics datasets were searched using Proteome Discoverer (Ver. 3.01.27). Data were processed using the ProSightPD high/high cRAWler node utilizing the Xtract algorithm and searched using the Proteome Discoverer ProsightPD4.2 annotated proteoform and subsequence search nodes. For the Annotated proteoform a precursor tolerance of 1.1 Da and a fragment mass tolerance of 10 ppm, methionine truncation, N-terminal formylation, and acetylation were allowed, a minimum of three fragments, while for the subsequence search, precursor and fragment tolerances were set to 10 ppm, minimum six fragments. To allow identification of disulphide bridges the searches were run in duplicate with fixed dehydro cysteine included or excluded, respectively. Only proteoforms with an FDR < 1% are reported.

All re-processed LC-MS data have been deposited to the ProteomeXchange Consortium via the PRIDE partner repository with the dataset identifier PXD045039[97].

## Differential expression analysis

A comprehensive differential transcription and translation analysis was performed by using three different tools xtail[98], riborex[99] and deltaTE[100]. These tools utilizes DESeq2[101] in assessing the expression of data derived from both Ribo-seq and RNA-seq libraries.

To ensure the quality of our data, a Principal Component Analysis (PCA) was conducted prior to employing deltaTE (see Supplementary Fig. 12). The raw read counts obtained from Ribo-seq and RNA-seq libraries were normalized using DESeq2. Replicates corresponding to a specific growth condition were anticipated to form dense clusters, indicating similar variance among them.

Subsequently, we proceeded to run deltaTE using the raw read counts from all our libraries. The differential transcription of genes was inferred from the RNA-seq data, whereas the differential translation was inferred from the Ribo-seq data. The output from deltaTE was derived from DESeq2 tool, which utilized two-sided Wald tests based on negative binomial generalized linear models and employed Benjamini-Hochberg method for multiple comparison adjustment.

To refine the differential expression results, we set an adjusted p-value cut-off of 0.05 and a log2 fold change cut-off of 1, each applied separately to each library type. Notably, the log2 fold change cut-off was imposed on the shrunken log2 fold change values to enhance the accuracy of our results.

## Functional enrichment analysis

We carried out a functional enrichment analysis by utilizing the tool clusterProfiler[102] and conducted both an overrepresentation analysis (ORA) and a gene set enrichment analysis (GSEA). Our ORA employed a prefiltered gene list using a log2 fold change threshold (<−1

downregulated, > 1 upregulated respectively) and an adjusted p-value cut-off (< 0.05). To determine the log2 fold change and p-values for the Ribo-seq and RNA-seq libraries, we utilized the differential expression analysis tool deltaTE[100]. ORA is based on two-sided Hypergeometric test with multiple comparison adjustment computed by Benjamini-Hochberg method. For the GSEA, we used an unfiltered list of shrunken log2 fold change values as input. GSEA is based on two-sided weighted Kolmogorov-Smirnov-like statistic with multiple comparison adjustment computed by Benjamini-Hochberg method.

Performing a functional analysis requires an annotation database, which encapsulates Gene Ontology (GO) terms[103,104] for a given organism. This database describes the relationships between the organism's genes and the GO terms associated with biological processes, molecular functions, and cellular components. While such databases are plentiful for eukaryotes, prokaryotic annotation databases for most organisms are not as readily available.

Therefore, we generated a custom annotation database for *M. mazei* (see https://github.com/RickGelhausen/pathsnake/tree/master/databases/org.Mmazei.eg.db). This started with downloading the available annotation from UniprotKB[105], which provided identifiers for our database and the GO terms for each annotated protein. Following this, we filtered the data to only include entries for our specific strain. The final SQLite database was subsequently assembled using the AnnotationForge package[106]. The resulting tables were subsequently analyzed and the most significant results were plotted using ggplot package[107] in the R environment.

## Plasmid construction

The SPA-tag used in this study was amplified by PCR from plasmid pLH7.3 (Supplementary Table 2) and TA-cloned into pCRII TOPO (Invitrogen), yielding plasmid pRS1612. SPA-tag was extracted from pRS1612 by restriction with *BamHI* and *NotI* and the fragment purified by gel extraction. The purified fragment was ligated into *BamHI* and *NotI* linearized shuttle vector pRS1595[108], generating plasmid pRS1624.

Selected sORFs exhibiting different TE values and genomic location, were amplified from chromosomal *M. mazei* Gö1 DNA with respective primers (Supplementary Table 3). For cloning of the sORFs with their respective native promoter, the promoter region around 200 bp upstream of the predicted start codon was amplified together with sORF. The PCR products were intermediate TA-cloned into pCRII TOPO followed by restriction with *XhoI* and *BamHI* and ligation into *BamHI* and *NotI* linearized pRS1624. Fragments isolated from plasmids were purified with gel extraction before ligation. The resulting plasmids were confirmed by sequencing and are described in Supplementary Table 2.

For constructions of sORFs under control of P*mcrB* and *mcrB* RBS[109], sORFs were PCR amplified with the depicted primer. Restriction of PCR products was performed with *NotI* and *BamHI* followed by ligation into linearized pRS893. *NotI* restriction site was removed by site directed mutagenesis or mung bean nuclease treatment. sORFs with fused P*mcrB* and RBS were extracted from plasmids by restriction with *XhoI* and *BamHI*, purified by gel extraction and ligated into linearized pRS1624.

Plasmids were transformed into *M. mazei*\* by liposome mediated transformation[29,30]. Plasmid maps are shown exemplarily in Supplementary Fig. 13.

## Western blot

*M. mazei* cultures growing exponentially in 50 ml under nitrogen sufficient ( + N) or limiting (-N) conditions were harvested by centrifugation (3220 x g, for 30 min at 4 °C) at $OD_{600}$ = 0.4 ( + N) and $OD_{600}$ = 0.2 (-N). The obtained cell pellet was resuspended in approximately 1 ml media followed by centrifugation (16,100 x g, 5 min at 4 °C).

The pellet was resuspended in 400 μl buffer containing 50 mM Tris pH 7.6 and lysed with a GenoGrinder (SPEX CertiPrep, Metuchen, USA) at 1600 strokes for 2 ×3 min followed by 30 min centrifugation at 16,100 x g and 4 °C. The protein concentration of the cell extract (supernatant) was measured with the Qubit™ Protein BR Assay Kit (Thermo Scientific™, Waltham, Massachusetts, USA) on the Qubit™ 4 Fluorometer (Thermo Scientific™, Waltham, Massachusetts, USA). 30 μg of each cell extract was mixed with LDS-sample buffer (Invitrogen, Darmstadt, Germany) and was separated on a NuPAGE™ Bis-Tris gel, 12% (Invitrogen, Darmstadt, Germany) at 200 V for 35 min followed by transfer on a PVDF (BioRad, München, Deutschland) or Amersham™ Protran® nitrocellulose (Cytiva ™, Marlborough, Massachusetts, USA) membrane with TransBlot Turbo (BioRad, München, Deutschland). As a control, an equally loaded SDS PAGEs was performed in parallel, which was stained with Coomassie Brilliant blue (Carl Roth, Karlsruhe, Germany) as loading control and the other used for subsequent western blot analysis. For detection, monoclonal Antibody against the SPA-tag (FG4R) (Invitrogen, Darmstadt, Germany, catalog number #MA1-91878) and the second antibody Goat Anti-Mouse IgG (H + L)-HRP Conjugate (BioRad, München, Deutschland, catalog number #1706516) were used in a 1:10,000 dilution and visualized with SuperSignal™ West Femto Maximum Sensitivity Substrate (Thermo Scientific™, Waltham, Massachusetts, USA).

For membrane isolation, *M. mazei* cultures were grown in 1 l media until exponential phase and harvested by centrifugation (6371 x g, for 30 min at 4 °C). Cells were resuspended in 16 ml 50 mM Tris pH 7.5 and disrupted using a French press cell (Spectronic Unicam, Cambridge, UK) at $4135 \times 10^6$ N · m$^{-2}$ followed by centrifugation (7741 x g, for 30 min at 4 °C). The supernatant was fractionated by additional ultracentrifugation (208,000 x g, for 1 h at 4 °C). Proteins were extracted from the membrane fraction by resuspending the 208,000 g pellet in 4 ml 50 mM Tris pH 7.5 supplemented with 2% Dodecyl-β-D-maltosid (DDM) followed by centrifugation (208,000 x g, for 1 h at 4 °C). Fractions were used for western blot analysis.

### Growth experiments
*M. mazei* overproduction strains were inoculated in 50 ml minimal media supplemented with puromycin (5 μg/ml) either under +N or -N condition from a preculture at mid exponential phase. The growth was monitored by measurement of OD$_{600}$ over the time.

### Conservation analysis
Unannotated small protein homologues identification was performed by using blastp and tBlastn searches in archaea using the National Center for Biotechnology Information (NCBI) database (https://blast.ncbi.nlm.nih.gov/Blast.cgi). As query sequences, the protein sequences for unannotated protein candidates identified by Ribo-seq were used. For tBlastn, the following parameters were used: the filter for low complexity regions off, a seed length that initiates an alignment (word size) of 6, 80% coverage of the query sequence with at least 60% identity, an E-value (Expect value) of ≤ 100 with an E-value between 0.01 up to 1 for high confidence hits[110]. Moreover, unannotated small proteins discovered in this study were further analysed for secondary structure and predicted protein domains, predictions of lipoproteins, as well as potential subcellular localization using predictions from the Phyre2 v2.0 (http://www.sbg.bio.ic.ac.uk/~phyre2/), SignalP-6.0 (https://services.healthtech.dtu.dk/services/SignalP-6.0/) TMHMM v2.0 (https://services.healthtech.dtu.dk/service.php?TMHMM-2.0) and PSORTb v3.0.3 servers (https://www.psort.org/psortb/)[111]. For selected small proteins, the protein structure was predicted by AlphaFold Protein structure database (https://alphafold.com/)[59,60].

### Statistics
To ensure the quality of our data for differential expression analysis, a Principal Component Analysis (PCA) was conducted prior to employing deltaTE (see Supplementary Fig. 12). For differential expression analysis in Figs. 3D, E, F, 4D, E, 5D, E, Supplementary Fig. 10A, B, the output from deltaTE was derived from DESeq2 tool, which utilized two-sided Wald tests based on negative binomial generalized linear models and employed Benjamini-Hochberg method for multiple comparison adjustment. To refine the differential expression results, we set an adjusted p-value cut-off of 0.05 and a log2 fold change cut-off of 1, each applied separately to each library type. Notably, the log2 fold change cut-off was imposed on the shrunken log2 fold change values to enhance the accuracy of our results. A Pearson correlation coefficient was computed to assess the linear relationship between global RIBO-seq log$_2$Foldchange and RNA-seq log$_2$Foldchange values differentially expressed genes as shown in Fig. 3D. For functional enrichment of differentially transcribed and translated genes shown in Fig. 3G, H and Supplementary Fig. 1, we conducted overrepresentation analysis (ORA) and a gene set enrichment analysis (GSEA). ORA is based on two-sided Hypergeometric test with multiple comparison adjustment computed by Benjamini-Hochberg method. For the GSEA, we used an unfiltered list of shrunken log2 fold change values as input. GSEA is based on two-sided weighted Kolmogorov-Smirnov-like statistic with multiple comparison adjustment computed by Benjamini-Hochberg method. Mean and standard deviation were computed for Figs. 3C, 7A, B, and Supplementary Figs. 2, 3B, and 10B.

### Reporting summary
Further information on research design is available in the Nature Portfolio Reporting Summary linked to this article.

### Data availability
Ribo-seq and RNA-seq data generated and analyzed during the current study have been deposited in GEO with the accession number GSE240615. The Ribo-seq for *M. mazei* can be viewed with an interactive online JBrowse instance (http://www.bioinf.uni-freiburg.de/ribobase) on request. All re-processed LC-MS data have been deposited to the ProteomeXchange Consortium via the PRIDE partner repository with the dataset identifier PXD045039, the meta-data is provided in Supplementary Data 10. This consists of previously published raw-files from the datasets PXD004325[56] [https://proteomecentral.proteomexchange.org/cgi/GetDataset?ID=PXD004325], PXD019792[54] [https://proteomecentral.proteomexchange.org/cgi/GetDataset?ID=PXD019792], PXD011996[55] [https://proteomecentral.proteomexchange.org/cgi/GetDataset?ID=PXD011996], as well as datasets PXD055745, and PXD055748, which were produced in house following the methodologies detailed in this manuscript, with publications pending. Source data are provided with this paper.

### Code availability
All the codes regarding functional enrichment analysis are available at the following GitHub repository: https://github.com/RickGelhausen/pathsnake. An archived version of the repository has been generated and is accessible via Zenodo with the following https://doi.org/10.5281/zenodo.13384255.

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

## Acknowledgements

We thank Stephanie Färber and Philipp Kible for technical assistance during the establishment of *M. mazei* Ribo-seq and Rebecca Eulitz during the in vivo validation. We further thank Eva Herdering for providing *M. mazei* cells grown under different stress conditions for LC-MS/MS analysis. This work was supported by the German Research Foundation (DFG) priority program SPP2002 "Small Proteins in Prokaryotes, an Unexplored World" (Grants SH580/7-1 and SH580/7-2 to C.M.S., CIBSS-EXC-2189-390939984 and BA2168/21-2 to R.B., TH 872/10-1 and 10-2 to A.T., RSCHM1052/20-1 + 20-2, RSCHM1052/19-2 to R.A.S.). This work was supported by the BMBF-funded de.NBI Cloud within the German Network for Bioinformatics Infrastructure (de.NBI) (031A532B, 031A533A, 031A533B, 031A534A, 031A535A, 031A537A, 031A537B, 031A537C, 031A537D, 031A538A).

## Author contributions

M.A.T., B.J., and L.H. contributed equally and are co-sharing first authors. C.M.S., R.B., and R.A.S. initiated the project. L.H., C.M.S., and R.A.S. designed the experiments. B.J. and M.G. cultured *M. mazei* and harvested cell pellets for Ribo-seq under L.H. supervision. L.H. established Ribo-seq for *M. mazei*. B.J., M.A.T., and L.H. manually curated the predicted unannotated sORFs, the annotated sRNAs, suspected mis-annotated genes, and consolidated the data from different genome annotations including the newest annotation. B.J. performed the conservation analyses, predicted the function of unannotated small proteins, and performed the cloning and western blot analyses together with L.He. R.G. performed the bioinformatic processing of the Ribo-seq data, established a workflow for differential expression analysis for Ribo-seq data, created an annotation database for *M. mazei* and developed a script for functional enrichment analysis. M.A.T. further analysed the differential expression data and did functional enrichment analysis and visualization. B.J. and M.A.T. did the data visualization under the supervision of R.A.S. T.H. conducted the structure prediction analysis with AlphaFold 2. L.H. and R.G. created a JBrowse instance for the described Ribo-seq datasets in the RIBOBASE database to facilitate their accessibility to the scientific community. L.C. and A.T. performed all LC-MS/MS analyses including interpretation. B.J., M.A.T., and L.H. wrote a first draft of the manuscript. B.J., M.A.T., L.He. and R.A.S. finalized writing the manuscript. B.J., M.A.T., and R.A.S. incorporated the changes according to the feedback from all the authors. R.B., C.M.S., and R.A.S. supervised the research and provided resources and funding. All authors approved the submitted version.

## Funding

## Competing interests

The authors declare no competing interests.
