## [Peer Review File · Nature Communications]

REVIEWER COMMENTS

Reviewer #1 (Remarks to the Author):

The manuscript „Exploring the small proteome of *Methanosarcina mazei* ...” by Jordan et al. describes a very thorough study about gene expression in this methanogenic archaeon. The transcriptome was analyzed by RNA-Seq, and in parallel the translome was analyzed by ribosomal profiling (Ribo-Seq). These analyses were performed under two different conditions, nitrogen sufficiency (+N) and nitrogen starvation (-N). In contrast to other RNA-Seq/Ribo-Seq studies, this study focused on the identification of novel small proteins of less than 70 aa (called sORFs), which had not been identified and annotated with other approaches previously. Noteworthy, the bioinformatics predictions were followed by manual curation, leading to a high-fidelity set of novel translated sORFs.

To complement these approaches, the results of past proteome studies were re-analyzed by using the newest genome annotation and including the novel sORFs. Translation of 62 previously annotated and 28 novel sORFs could be verified.

Furthermore, 15 sORFs were chosen, cloned into an expression vector, produced as fusion proteins with an affinity tag, and 12 proteins could be detected by Western blotting, underscoring their existence as bona fide small proteins. In a last approach it was shown that the overproduction of two of these small proteins resulted in a growth defect either under +N or under -N condition, indicating that they are important for (the regulation of) nitrogen metabolism.

Taken together, the manuscript describes the most extensive multi-approach study of gene expression/sORF identification performed with an archaeon, and I congratulate the authors to this large set of results.

Nevertheless, I have a various points that should be addressed prior to publication.

- 1) Line 81. Please add a few references after “diverse organisms”
- 2) L. 88. In my opinion the bracket should be removed, because MNase also trims translated regions with the exception of about 30 protected nt. Only the sum of all ribosomes enables the differentiation of translated and untranslated regions.
- 3) L. 98 (and other places). “Proteins” are not “translated”, but transcripts.
- 4) L. 220ff. I find the manual curation as well as the chosen criteria very good.
- 5) L. 267. What does “SEP” mean? Maybe a list of abbreviations would be helpful.
- 6) L. 315. Is this custom annotation database available?
- 7) L. 371. Puromycin is a translation inhibitor, why was it added during growth?

8) L. 394. “Fig. 1A” does not show the specific optimized Ribo-seq protocol, therefore, I propose to remove the brackets at this place.

9) L. 398. What does “rapid cooling” mean, please give some information about the time.

10) L. 403/404. Figures 1C and D do not show “ribosome footprints”.

11) Fig. 1B. Even if the information is (very small) included, I propose to add “+N” and “-N” on top of the two Figures. In the legend, only the “green profile” is explained (which looks blue to me), but not the “red profile”.

12) L. 472. Small N-terminal extensions/truncations can probably not be resolved by Western blotting, I propose to remove line 472.

13) e.g. Fig. 3 C, E, F (+ other Figures). The Figures include text that is too small to be read. I propose to change the size or to remove the text. Fig. 3E. The y-axis has no label.

14) L. 537. Add “to be translated” after “predicted”.

15) Fig. 4G. 3 small ribosomal proteins are shown to be down-regulated. It might be good to add whether or not the other ribosomal proteins are also down-regulated.

16) L. 627f. When the proteins do not contain a signal sequence for secretion, what were the criteria to predict that they act extracellularly?

17) L. 633. 76% of the proteins have the canonical ATG/GTG start codon, therefore, there must be other reasons why they have been overlooked in previous studies.

18) L. 641. Please explain the criteria for the prediction of “operon structures”.

19) L. 649. I propose to start the line with “(E, F)” to indicate that the following text does not belong to (D).

20) L. 659-681. I am confused. “15 candidates were selected”, but only 13 were “episomally expressed”, and then Fig. 4A/B show Western results for 15 proteins.

2 are “internal to annotated genes” and are “thus not included in the list of 314 novel ORFs”. However, Figure 5C includes the category “overlapping with gene” and the two proteins carry the designations “sORF_03/_10”, like the other novel proteins.

Translation of 10 sORFs was validated in Fig. 6A, “the remaining” (line 669) according to the Figure must be sORFs_12/_13/_16, not _8/_9/_15.

21) L. 713. Please change “16” to “15”.

22) L. 728ff. I got the expression that the growth experiments were not performed with “mutant strains”, but with the wildtype containing expression plasmids.

23) Discussion. In my opinion the Discussion is MUCH too long and too unfocused. The specific power of the “multiomic strategy” is repeated at many places. There are rather long repetitions from the Introduction and the Results section. In my opinion the effect of the Discussion could be highly enhanced when it would be drastically shortened.

- 24) L. 778-784. This is a very long sentence for the English language.
- 25) L. 799. The translation efficiency is determined by the frequency of initiation, not the speed of the ribosomes.
- 26) L. 808. Non-canonical AUG?
- 27) L. 824ff. First it is argued that DeepRibo predicted 37% more sORFs, and, therefore, it was chosen, because a high number of predictions is good. Then it is argued that DeepRibo generates a high rate of false positive predictions and needs manual curation to eliminate these predictions. Taken together, is DeepRibo really better than REPARATION?
- 28) L. 894. (Fig. 2B) does not show 51 ORFs with truncated N-termini, but just one example.
- 29) L. 894/891. 47% and “the remaining 17%” do not add up to 100%.
- 30) L. 902-907. It is true that TIS-Seq would increase the prediction of initiation sites, however, TIS-Seq was not performed.
- 31) L. 915ff. This is true, but was already part of the Introduction.
- 32) L. 971. Are 300 more Western blots really “vital” for future research?
- 33) L. 1012f. This is a lot of speculation based on a negative result.
- 34) Fig. 8A. In my opinion the rather long discussion about sRNA154 and Fig. 8A distract from the strength of the study, i.e. giving a genome wide overview with Ribo-Seq + RNA-Seq.
- 35) Fig. 8B. I do not really understand the Figure. “Manual curation” encompasses all other methods, it is shown to lead to 289 predictions, which is in contrast to the 314 predictions given in the Figure legend and the text. When manual curation led to 289 predictions, DeepRibo only to 73 (none of them exclusive to DeepRibo), was DeepRibo needed at all? In addition, the number 73 is in contrast to 266 after strict filtering (L. 607) and 63 after filtering and manual curation (L. 611). In the text it is stated that 28 novel sORFs were verified by bottom up proteomics (L. 686) and 3 additional by top-down proteomics (L. 688), which does not add up to the 26 shown in Fig. 8B.
- 36) L. 458. Replace “annotated ORF” with “annotated start codon”.
- 37) Even if the manuscript has a focus on small proteins, I suggest to add information that is typical for Ribo-Seq studies, e.g. the length of the protected fragments (in comparison to bacterial and eukaryotic ribosomes), a possible enrichment on start/stop codons or pause sites, the presence of a translation initiation motif, etc.

Reviewer #2 (Remarks to the Author):

The paper of Joran et al presents Ribo-seq and RNA-seq analysis of the translome and transcriptome of the archaeon *Methanosarcina mazei* under nitrogen-rich and nitrogen-depleted conditions. On the bases of these data, the authors predict 314 new short ORFs in addition to 93 previously known. They verified that some of these ORFs are indeed expressed in the cell.

Critique

This is a technically sound study, which, however, does not provide major new revelations. As the authors accurately summarize at the end of the Introduction section: they provide “a large catalogue of novel *M. mazei* sORFs”. As a resource for *M. mazei* aficionados, this is definitely a very important study. But for a more general scientific community it is less useful because the catalogues of sORFs and small proteins have been described for many bacterial and also archaeal organisms. Because of that, in the opinion of this reviewer, this paper would be an excellent candidate for a more specialized journal.

The paper is clearly written but is unnecessarily long and superfluous. Several sections of the Results section belong to the Methods section. Some parts can be deleted without sacrificing the message. Discussion can be significantly shortened.

Here are just few examples:

ll. 390-410. The first two paragraphs of Results are purely technical and belong to the Methods.

ll. 442-480. The entire section 3.2 (Insights into protein translation in *M. mazei* through Ribo-seq), including Figure 2, can be eliminated because similar observations (misannotated start codons, truncated or extended genes, etc.) have been described for many genomes.

Too many experimental details spill into Results which, even if counterintuitive, makes reading less informative. For example, Fig. 4A shows the results of sORF prediction by three different methods (!). The paper would benefit if all these go to the Methods section and the readers are simply presented with the most reliable findings. In fact, this entire section (ll. 603-618) can be replaced with just its last sentence: “We can provide a list of 314 small novel proteins with high confidence.”

Reviewer #3 (Remarks to the Author):

In the paper called “Exploring the secret small proteome of *Methanosarcina mazei* through Ribo-seq and peptidomics, uncovering a multitude of dual-function RNAs” by Britta Jordan^{1*}, Muhammad Aammar Tufail^{1*}, Lydia Hadjeras^{2*}, Rick Gelhausen³, Liam Cassidy⁴, Tim Habenicht¹, Miriam Gutt¹, Rolf Backofen³, Andreas Tholey⁴, Cynthia M. Sharma², Ruth A. Schmitz^{1,#}, a Ribo-seq protocol is established for the methanogenic archaeon *Methanosarcina mazei*. This tool provides valuable insights into protein synthesis (e.g. identification of novel coding ORFs and validation of annotated start codons). Ribo-seq was further employed to compare the translome under standard growth conditions and nutrient limitation, with a particular focus on small proteins (< 70 aa). A variety of novel sORFs were predicted and certain candidate sORFs were put forward which possibly play a role in nitrogen metabolism based on phenotypical data. This paper is nicely written and provides a wealth of information. Results are discussed in a pleasant and detailed way with a good number of supporting figures. I appreciate the author’s work in confirming the ribo-seq data using mass spec and western blotting. Given that ribo-seq is still its infancy for archaeal microorganisms, I am convinced that this manuscript is highly valuable for other researchers to further expand this technique in this domain of life. My major scientific remark is related to the in vivo analysis of tagged sORFs, where I do not fully support the decisions on episomal overexpression of the sORFs. In addition, I believe that the lay-out and order of the (sub)figures should be improved to get rid of the ‘messy’ appearance of certain figures.

Major remarks:

1) “Section 3.5.4 Targeted and untargeted in vivo validation of selected candidates”: In this section, it is described that candidate sORFs are selected based on their genomic context (L661) and that these sORFs were episomally expressed in *M. mazei* under control of the constitutive PmcrB promoter and a standard *M. mazei* ribosome binding site and fused to a C-terminal sequential affinity tag (SPA) to assess their in vivo translation (L663-665). I believe this in vivo validation of tagged sORF is a nice contribution to the paper. However, I do not understand the reasoning behind selection of a constitutive PmcrB promoter and a standard *M. mazei* ribosome binding site for driving sORF expression as a first screening in Fig 6A, given that this deviates from the endogenous genomic context of the sORFs (this is not in line with why these candidates were selected in the first place). Based on this general screening, three candidate sORFs were selected to study using their endogenous promoter elements (Fig 6B). However, I believe the latter approach using the native promoter/RBS is more valuable than the analysis using a non-related constitutive promoter. I would furthermore argue that results could be different when using the native promoter elements of the sORF compared to the PmcrB promoter and a standard *M. mazei* ribosome binding site (on plasmid). I would advise to also confirm translation of the other 10 sORFs in the set-up employing their endogenous promoter element. Perhaps some of these other sORFs also display a differential

expression in N conditions? In the same line, I wonder why insertion of the epitope tag into the genome was not the preferred option here, as this would be the closest to the natural expression of the ORF (and not overexpress the sORF on plasmid)?

2) The lay-out of nearly all figures should be improved. The text size in many figures is too small and barely readable, e.g. Fig 3 C-H, Fig 4 B-H, Fig 5E-G. Please also take care to use the same (or at least similar) text fonts, text sizes and color-schemes in different subfigures to avoid the figures looking 'messy' / 'cluttered' (e.g. Fig 3/4/5).

3) In the text, Fig 4A, 4C and 4D are mentioned before Fig 4B. Also for Fig 5 and Fig 7, the order in the text is non-chronological. Please rearrange the labeling of the subfigures.

Minor remarks:

General:

1) For certain products, the specifications and manufacturer are missing in the M&M section (e.g. "M-MLV reverse transcriptase" at L 159, "a high-fidelity DNA polymerase" at L162)

2) What is the size of the generated footprints? Unless I missed this, I could not find this information in the manuscript.

3) Did the authors provide the log-in details to access the generated Ribo-seq data on <http://www.bioinf.uni-freiburg.de/ribobase> ?

4) There seems to be an issue with the doi-links in the reference list ("https://doi.org:https://doi.org")

5) A short section on the (technical) challenges that must be overcome on the implementation of Ribo-seq for anaerobic microorganisms could be beneficial to allow the reader to assess the feasibility of implementing this technique for other archaea (e.g. thermophiles, acidophiles,...).

6) L353 "The supernatant was resuspended in LDS-sample buffer (Invitrogen, Darmstadt, Germany) and separated on a NuPAGE™ Bis-Tris gel, 12% (Invitrogen, Darmstadt, Germany) at 200 V for 35 min". How much total protein was loaded on gel? For some blots, this is mentioned in the figure's legend, but not for all. Please provide this information for all blots.

Specific:

7) L51-52 "Small proteins in the domain of archaea are underrepresented in recent studies, and only few small proteins are characterized until now 12-14". The references correspond to small proteins in Euryarchaeotes only. Is it correct to assume that no small proteins have been characterized in other archaeal phyla?

8) L194 “To ensure the accuracy of the predictions, a rigorous filtering process was performed and manual curation of the predicted ORFs was carried out”. Could you elaborate on which sequences were filtered and how this was performed?

9) L413-418 “Conversely, the sRNA162, a known non-coding regulatory sRNA 87, displayed high cDNA read coverage exclusively in the RNA-seq library (Fig. 1C right panel). This observation further validates its non-coding nature. Furthermore, the read coverages of the 5'- and 3'-UTRs of psmB demonstrated higher levels in the RNA-seq library compared to the Ribo-seq library. This indicates that mRNA regions that are not translated”. How can you explain the presence of (low number of) reads detected in the Ribo-seq for these RNA (especially for sRNA162)? Would this just be background? If so, did you set a threshold for the number of reads below which a non-coding nature could be attributed to? If so, perhaps this could be indicated (as e.g. a dotted line) on the graph?

10) L416 “Furthermore, the read coverages of the 5'- and 3'-UTRs of psmB demonstrated higher levels in the RNA-seq library compared to the Ribo-seq library.” A reference to “(Fig. 1C, left panel)” could be added to make it more clear.

11) L483 “M. mazei genome has a total of 3,440 annotated ORFs. DeepRibo algorithm predicted 1,566 and 1,430 annotated ORFs under +N and -N conditions, respectively, overall detecting 47% of all annotated ORFs (Fig. 3A).” How does this compare to MS coverage?

12) L489: “Analysis of the start and stop codon distribution of annotated ORFs (Fig. 3B) shows that M. mazei favours the canonical start ATG by 79%. The two mainly used stop codons are TAA (53%) and TGA (42%).”. Are the rarest start and stop codons overrepresented in a particular COG-category? Is there a correlation between start and stop codon usage and a COG-category?

13) L525 “Among the three most enriched Gene Ontology (GO) terms for biological processes under N limited conditions were ‘nitrogen fixation, ‘nitrogen cycle metabolic process’, and ‘regulation of nitrogen utilization’ (Fig. 3G).”. I believe it should be worth noting that the response of ‘nitrogen fixation’ and ‘nitrogen cycle metabolic process’ is way more pronounced than ‘regulation of nitrogen utilization’ (the latter is very comparable to the other groups in the top 10), according to Fig 3G.

14) L581: “Under -N, 5 sORFs were found to be upregulated and 11 sORFs downregulated at the RNA level (as seen in Fig. 4F). On the other hand, at the RIBO level, 9 sORFs were observed to be upregulated while 22 sORFs were downregulated (depicted in Fig. 4G), highlighting the complex regulatory mechanisms at work in gene expression.” Could you speculate some more on what could be happening here?

15) L617 “After finalizing the analysis, we can provide a list of 314 small novel proteins with high confidence.” A reference to Table S8 at this point would be helpful.

16) You mention at L608 that sORF detection is difficult in long genes. However at L635 “The 314 novel sORFs are encoded in different genomic context (Fig. 5C)[...]”. You do compare the genomic regions of the 314 novel sORFs. This makes me wonder whether you could give a prediction/estimation on how many sORFs could potentially still be present in the long genes and you are “missing” here?

17) L667 “Out of those 13 candidates, the translation of 10 sORFs was validated in a western blot analysis using a FLAG directed antibody against the SPA-tag (see Fig. 6A). “ A lot of smearing/different sized bands are observed in the blot for sORF_07. How could this be explained?

18) L728 “We further performed growth experiments with all *M. mazei* mutant strains episomally expressing sORFs. Most of those mutants did not show any phenotype compared to the empty vector control (data not shown).” Under control of which type of promoter/RBS? I would still advise to add the growth curve of all other mutants to the supplementary data of this paper for transparency.

Re: Point-to-point reply to the reviewers for manuscript: NCOMMS-23-48028

Thank you very much for supervising the review process of our manuscript. We greatly appreciate the reviewers' time and expertise, which have significantly contributed to enhancing the quality and clarity of our manuscript. We further thank the reviewer for the supportive judgement as the 'most extensive multi-approach study of gene expression/sORF identification performed with an archaeon and congratulate the authors to this large set of results' and being 'highly valuable for other researchers to further expand this technique in this domain of life.' We have addressed all the comments carefully, performed additional experiments and have made appropriate revisions, which we believe have strengthened the paper.

Following are the comment wise responses to the questions or comments of worthy reviewers. We have made our best attempt to address their valuable recommendations.

Note: All the changes are highlighted in **red colour** in the revised manuscript throughout the text and the response to the reviewers are highlighted in **blue in the following point by point response**.

Reviewer #1 (Remarks to the Author):

The manuscript „Exploring the small proteome of Methanosarcina mazei ...” by Jordan et al. describes a very thorough study about gene expression in this methanogenic archaeon. The transcriptome was analyzed by RNA-Seq, and in parallel the translome was analyzed by ribosomal profiling (Ribo-Seq). These analyses were performed under two different conditions, nitrogen sufficiency (+N) and nitrogen starvation (-N). In contrast to other RNA-Seq/Ribo-Seq studies, this study focused on the identification of novel small proteins of less than 70 aa (called sORFs), which had not been identified and annotated with other approaches previously.

Noteworthy, the bioinformatics predictions were followed by manual curation, leading to a high-fidelity set of novel translated sORFs.

To complement these approaches, the results of past proteome studies were re-analyzed by using the newest genome annotation and including the novel sORFs. Translation of 62 previously annotated and 28 novel sORFs could be verified.

Furthermore, 15 sORFs were chosen, cloned into an expression vector, produced as fusion proteins with an affinity tag, and 12 proteins could be detected by Western blotting, underscoring their existence as bona fide small proteins. In a last approach it was shown that the overproduction of two of these small proteins resulted in a growth defect either under +N or under -N condition, indicating that they are important for (the regulation of) nitrogen metabolism.

Taken together, the manuscript describes the most extensive multi-approach study of gene expression/sORF identification performed with an archaeon, and I congratulate the authors to this large set of results.

Nevertheless, I have a various points that should be addressed prior to publication.

1)Line 81. Please add a few references after “diverse organisms”

We added new references accordingly (line 85-86).

2)L. 88. In my opinion the bracket should be removed, because MNase also trims translated regions with the exception of about 30 protected nt. Only the sum of all ribosomes enables the differentiation of translated and untranslated regions.

We agree with the reviewer and removed these brackets (line 93).

3)L. 98 (and other places). “Proteins” are not “translated”, but transcripts.

We fully agree with the reviewer. To address this, we have gone through the manuscript and corrected the usage of the word translated (essentially removing it when in context of a protein).

4)L. 220ff. I find the manual curation as well as the chosen criteria very

good. We greatly appreciate this feedback.

5)L. 267. What does “SEP” mean? Maybe a list of abbreviations would be helpful.

The term SEP is an abbreviation for **S**mall open-reading frame **E**ncoded **P**rotein - the protein produced following translation of an sORF. The term SEP is used as opposed to small protein, as small proteins may have been produced via posttranslational processing. We now avoid that term for clarity and continuously use the terms sORF (gene) or small protein. In addition, we now provide a list of abbreviations added in the main manuscript after the abstract as following:

Term	Abbreviation
Nitrogen sufficiency	+N
Nitrogen limitation	-N
Open reading frame	ORF
Small open reading frame	sORF
Amino acid	aa
Micrococcal nuclease	MNase
Dithiothreitol	DTT
Dodecyl- β -D-maltosid	DDM
Optical density	OD
Bottom-up proteomics	BUP
Top-down proteomics	TDP
Reads Per Kilobase per Million mapped reads	RPKM
False discovery rate	FDR
Translation efficiency	TE
Principal Component Analysis	PCA
log2 fold change	log2FC
Overrepresentation analysis	ORA
Gene set enrichment analysis	GSEA
Gene Ontology	GO
National Center for Biotechnology Information	NCBI
Transcriptional start site	TSS
Translation initiation site	TIS

6) L. 315. Is this custom annotation database available?

The custom annotation database is available on the GitHub repository that was provided during the submission process.

<https://github.com/RickGelhausen/pathsnake/tree/master/databases/org.Mmazei.eg.db>

We apologize that the annotation database was not referenced directly in the manuscript. We have now added a respective reference link in the manuscript and improved the structure of the GitHub repository in line 320.

7)L. 371. Puromycin is a translation inhibitor, why was it added during growth?

We apologize for not being explicit here. In this case, puromycin was used as a selection to keep the plasmid. We episomally expressed the *M. mazei* sORFs under a constitutive or native promoter from an extrachromosomal plasmid, for which puromycin resistance is used as a selection marker to be stably kept. Puromycin resistance is one of the few available selection markers for methanogenic archaea.

8)L. 394. “Fig. 1A” does not show the specific optimized Ribo-seq protocol, therefore, I propose to remove the brackets at this place.

We agree with the reviewer's comment, and we thus changed this sentence. In the revised manuscript, the sentence now reads as follows (line 398-402):

“To generate a translome map that provides a comprehensive depiction of translated annotated sORFs and to discover novel sORFs in *M. mazei* Gö1, we optimized the Ribo-seq protocol initially proposed by (Hadjeras et al. 2023a, Hadjeras et al. 2023b, Oh et al. 2011)^{40,48,52} and further developed it to suit the characteristic of this particular methanoarchaeon under two different growth conditions (+N and -N) (Fig. 1A and see Materials and Methods for more detailed information).

9)L. 398. What does “rapid cooling” mean, please give some information about the time.

Rapid cooling in this section refers to our “fast-chilling” harvest approach, which consists of rapid arrest of translation and stabilization of polysomes by fast cooling of the cultures in ice-water baths with gentle agitation for 3 min, followed directly by centrifugation to collect cells and snap-freezing in liquid nitrogen.

In the course of shortening the results section and deleting overly detailed method descriptions, we have completely deleted this section. However, the exact information was described and can still be found in the Material and Methods section. We now refer to the Methods section in the results.

0) L. 403/404. Figures 1C and D do not show “ribosome footprints”.

We apologize for not being correct here. We now changed the information concerning Fig. 1C and D to “translated regions of the genome”.

1) Fig. 1B. Even if the information is (very small) included, I propose to add “+N” and “-N” on top of the two Figures. In the legend, only the “green profile” is explained (which looks blue to me), but not the “red profile”.

We thank the reviewer for this feedback and added +N and -N to Fig. 1B. We increased the size of the conditions in Fig. 1A and explained both conditions in the figure legend. Moreover, we changed and optimized Figs 2-5 according to the two reviewer's comments.

2) L. 472. Small N-terminal extensions/truncations can probably not be resolved by Western blotting, I propose to remove line 472.

We agree with the reviewer, the changes in mol. mass are too small to be detectable by western blotting, we now removed that part as requested.

3) e.g. Fig. 3 C, E, F (+ other Figures). The Figures include text that is too small to be read. I propose to change the size or to remove the text. Fig. 3E. The y-axis has no label.

We thank the reviewer for this valuable comment. We have now thoroughly modified the figures 3C, E, F and other figures, the size of the most important text has been increased, and several less important text parts have been deleted. We have corrected the axis labels for Fig. 3E.

4) L. 537. Add "to be translated" after "predicted."

We agree with the reviewer and have added the information accordingly.

5) Fig. 4G. 3 small ribosomal proteins are shown to be down-regulated. It might be good to add whether or not the other ribosomal proteins are also down-regulated.

The point is well taken by the reviewer. The majority of the other ribosomal proteins are also down regulated and this has been mentioned in Table S6 of the original MS.

6) L. 627f. When the proteins do not contain a signal sequence for secretion, what were the criteria to predict that they act extracellularly?

To identify small proteins that are secreted over a membrane without a signal sequence, various mechanisms have been proposed in scientific research (Dupont et al., 2007 and Cohen et al., 2020). One key method involves the use of bioinformatics tools to predict potential secretory proteins based on specific features such as size, hydrophobicity, and secondary structure. Additionally, experimental techniques like mass spectrometry and proteomics have been instrumental in identifying unconventional secretion pathways for proteins lacking classical signal sequences. We used the bioinformatics tools mentioned in the Table S8. Mainly for subcellular localization, we used PSORTb v 3.0.3 servers (<https://www.psort.org/psortb/>), which utilizes the power of support vector machines (SVMs) and a sophisticated Bayesian network. These machine learning algorithms analyze protein sequences to predict their localization, leveraging patterns identified from an extensive and meticulously curated training dataset. Ensuring the accuracy and reliability of predictions, PSORTb 3.0 undergoes thorough validation

through 5-fold cross-validation and independent proteomics analysis. This rigorous evaluation process guarantees high precision and recall across a broad spectrum of prokaryotes, including those with unconventional membrane structures, thereby solidifying the robustness of the predictive model.

Overall, these are predictions for the localization of the small proteins. Indeed, it is attractive to evaluate and speculate how small proteins are at all identified to be secreted over a membrane, if a signal sequence is not present.

7) L. 633. 76% of the proteins have the canonical ATG/GTG start codon, therefore, there must be other reasons why they have been overlooked in previous studies.

Correct, only 24% have a non-canonical start. The most likely reason that the others with an ATG/GTG as start codon have not been identified based on genomic information in previous studies is the small size of those ORFs. Automated annotation programs in previous times often set the minimal length for an ORF to code for at least 100 amino acids. This is also mentioned in the text (introduction).

8) L. 641. Please explain the criteria for the prediction of “operon structures”.

To predict operon structures we used a tool called Operon-Mapper (https://biocomputo.ibt.unam.mx/operon_mapper/), which predicts operons in bacterial or archaeal genomes based on the intergenic distance of neighboring genes as well as the functional relationships of their protein-coding products (Taboada et al., 2018). The 9% of sORFs are indeed localized within genes of known or predicted operons - within the intergenic space. We now modified the wording respectively.

9) L. 649. I propose to start the line with “(E, F)” to indicate that the following text does not belong to (D).

We agree and have changed the figure legend now according to the reviewer’s suggestion.

10) L. 659-681. I am confused. “15 candidates were selected”, but only 13 were “episomally expressed”, and then Fig. 4A!B show Western results for 15 proteins. 2 are “internal to annotated genes” and are “thus not included in the list of 314 novel ORFs”. However, Figure 5C includes the category “overlapping with gene” and the two proteins carry the designations “sORF_03!_10”, like the other novel proteins. Translation of 10 sORFs was validated in Fig. 6A, “the remaining” (line 669) according to the Figure must be sORFs_12/_13/_16, not _8/_9/_15.

We agree with the reviewer that this paragraph is somewhat unclear regarding the number of selected sORFs. We have critically reviewed the number of selected sORFs and their names and changed the corresponding places in the text to make everything clear. 15 ORFs were selected for validation by western blot. Two of them (SORF3 and sORF10) are not from the list

of the new 314 sORFs, since they are located within a larger gene. 13 of these selected sORFs were expressed episomally with a constitutive promoter (including sORF8). Of these 13 sORFs, ten showed a positive result in the blot. The two remaining sORFs together with sORF8, which was also tested with a constitutive promoter, were expressed episomally under their native promoter.

Nevertheless, at the request of another reviewer, we examined the expression of further sORFs under their native promoter, which led to a overall change in the numbers. We have even included another sORF, namely sORF2, in the analysis and therefore now have a total of 16 selected sORFs for *in vivo* validation. From those 16 sORFs, 10 were cloned under their native promoter fused to a C-terminal SPA-tag. Here, we were able to detect 8 sORFs expressed under their respective native promoter, from which 3 (sORF3, sORF8 and sORF9) showed differential expression based on the nitrogen availability (see Fig. 6A). Due to methodological circumstances, not all 16 selected sORFs could be cloned under their native promoter. Thus, 13 of these 16 were expressed episomally with a constitutive promoter. Of these 13 sORFs, 10 showed a positive result in the western blot (see Fig. S5A). In summary, validation of sORF expression under either the native or a constitutive promoter via epitope tagging and immunoblotting analysis was successful for 13 out of 16 selected sORFs. However, the missing three sORFs were detected by LC-MS (Fig.6B).

Figure 5C contains only partially overlapping genes, which is not the case for sORF3 and 10. The latter sORFs are completely overlapping (internal sORFs). We have amended the description in figure 5C accordingly and agree with the reviewer that it was not entirely clear before.

11) L. 713. Please change “16” to “15”.

We apologize for the incorrect naming and have corrected it accordingly.

12) L. 728ff. I got the expression that the growth experiments were not performed with “mutant strains”, but with the wildtype containing expression plasmids.

We thank the reviewer for this important comment and changed it accordingly. We often call the wild type with an empty plasmid or on overexpression plasmid also a mutant to indicate the vector presence. However, the correct way is wild type containing the empty expression vector and overexpression strains. We now changed the wording in the manuscript accordingly.

13) Discussion. In my opinion the Discussion is MUCH too long and too unfocused. The specific power of the “multiomic strategy” is repeated at many places. There are rather long repetitions from the Introduction and the Results section. In my opinion the effect of the Discussion could be highly enhanced when it would be drastically shortened.

We thank the reviewer for this valuable comment and modified and shortened the discussion thoroughly. The discussion now avoids redundancy with the results and introduction part and the part on the dual function RNA (RNA154) was significantly reduced as suggested by the reviewer see comment 33)2.

14) L. 778-784. This is a very long sentence for the English language. This is correct, due to the discussion reduction the sentence is not any more present.

15) L. 799. The translation efficiency is determined by the frequency of initiation, not the speed of the ribosomes.

We agree with the reviewer and apologize for not being correct here. We now changed to the correct definition - frequency of initiation. In addition, we have modified and shortened the discussion thoroughly.

16) L. 808. Non-canonical AUG?

We apologize for the mistake and have corrected it to ATG. In addition, the sentence has been modified due to shortened discussion.

17) L. 824ff. First it is argued that DeepRibo predicted 37% more sORFs, and, therefore, it was chosen, because a high number of predictions is good. Then it is argued that DeepRibo generates a high rate of false positive predictions and needs manual curation to eliminate these predictions. Taken together, is DeepRibo really better than REPARATION?

DeepRibo was selected for its higher sensitivity, indicated by its ability to predict 37% more sORFs than REPARATION. This characteristic is particularly advantageous in exploratory phases of research where the primary objective is to capture the broadest possible spectrum of sORFs, including those that may be less abundantly expressed or have weaker signals in the dataset.

The acknowledged higher rate of false positives associated with DeepRibo is a recognized trade-off for its increased sensitivity. We contend that this does not diminish its utility, rather it necessitates a rigorous post-prediction validation process. For this study, we prioritized a comprehensive identification strategy, which is why DeepRibo's sensitivity made it the tool of choice. The subsequent manual curation and validation processes, including LC-MS and in vivo protein validations, are standard and necessary steps to ensure the accuracy of predictions from high-sensitivity tools like DeepRibo.

Therefore, when considering the research goals, DeepRibo proved to be the more suitable option over REPARATION. It allowed us to cast a wider net for potential sORFs, ensuring minimal false negatives. The subsequent curation was a necessary step to refine the dataset, which we believe is a justified effort in the quest to provide a more complete landscape of the small proteome of *M. mazei* in our study. Thus, the initial selection of DeepRibo is defended not solely on the number of sORFs predicted but on the basis of its alignment with the study's explorative and comprehensive objectives.

18) L. 894. (Fig. 2B) does not show 51 ORFs with truncated N-termini, but just one example.

We apologize for not being precise and changed accordingly to '(Fig. 2B depicting one example)'. In addition, we have modified and shortened the discussion thoroughly.

19) L. 894/891. 47% and "the remaining 17%" do not add up to 100%.

We apologize for not being correct. We now have modified the sentence and now used absolute numbers to be more clear.

20) L. 902-907. It is true that TIS-Seq would increase the prediction of initiation sites, however, TIS-Seq was not performed.

We agree with the reviewer. We intended to point out the next steps, which are not part of this manuscript but are important for the planned re-annotation of the genome. This information is still in the shortened discussion but is not present anymore in the introduction

21) L. 915ff. This is true, but was already part of the Introduction.

We agree and apologize for the redundancy. In the shortened discussion, this aspect is not mentioned any more.

22) L. 971. Are 300 more Western blots really "vital" for future research?

We agree with the reviewer. We wanted to point out that the validation should be performed for those selected novel sORFs which are further studied, e.g. for function. Furthermore, we did not want to give the impression to perform the western blots for all - only on demand. In the process of shortening the discussion, this part has been deleted.

23) L. 1012f. This is a lot of speculation based on a negative result.

Point well taken, we completely agree with the reviewer. This is a very speculative part, since we were excited to see the high number of dual function RNAs. However, it is not required for this report, and we now deleted the part and kept it very short concerning the comparison with the previous results and visible effects of post transcriptional regulation by sRNA154 (Table 1). Moreover, the complete discussion has been shortened and focused to streamline the message of this manuscript on the genome wide overview (see also comment 23).

24) Fig. 8A. In my opinion the rather long discussion about sRNA154 and Fig. 8A distract from the strength of the study, i.e. giving a genome wide overview with Ribo-Seq + RNA-Seq.

We agree with the reviewer, see above. We took out the figure from the main manuscript and now reduced the respective text accordingly and only show the structure of the sRNA154 with the location of the predicted sORF in the supplement, since the sRNA154 is the central RNA for regulation in N-metabolism. And changed the title accordingly.

25) Fig. 8B. I do not really understand the Figure. "Manual curation" encompasses all other methods, it is shown to lead to 289 predictions, which is in contrast to the 314 predictions given in the Figure legend and the text. When manual curation led to 289 predictions, DeepRibo only to 73 (none of them exclusive to DeepRibo), was DeepRibo needed at all? In addition, the number 73 is in contrast to 266 after strict filtering (L. 607) and 63 after filtering and manual curation (L. 611). In the text it is stated that 28 novel sORFs were verified by bottom up proteomics (L. 686) and 3 additional by top-down proteomics (L. 688), which does not add up to the 26 shown in Fig. 8B.

We apologize for not being completely clear on this matter. Upon reevaluation, we have redesigned and corrected all the numbers mentioned in Fig. 8.

The acknowledged higher rate of false positives associated with DeepRibo is a recognized trade-off for its increased sensitivity. We contend that this does not diminish its utility, rather it necessitates a rigorous post-prediction validation process as a base for manual curation. For this study, we prioritized a comprehensive identification strategy, which is why DeepRibo's sensitivity made it the tool of choice. The subsequent manual curation and validation processes, including LC-MS and in vivo protein validations, are standard and necessary steps to ensure the accuracy of predictions from high-sensitivity tools like DeepRibo.

In addition, the correct number of 26 detections via MS has been confirmed, with 23 via bottom-up proteomics, 19 of which were validated by top-down proteomics, and an additional 3 through top-down proteomics (□ all together 26).

26) L. 458. Replace "annotated ORF" with "annotated start codon".

We agree with the reviewer and have modified it, respectively (line 450).

27) Even if the manuscript has a focus on small proteins, I suggest to add information that is typical for Ribo-Seq studies, e.g. the length of the protected fragments (in comparison to bacterial and eukaryotic ribosomes), a possible enrichment on start/stop codons or pause sites, the presence of a translation initiation motif, etc.

As recommended, we have incorporated the length distribution of the footprints retrieved in our study into panel **A** of a novel Supplementary Fig. S9. Additionally, we have included a sentence in the text (line 847-849) to address this plot: "Our Ribo-seq libraries, obtained under both nitrogen + and - conditions, exhibit a wide range of footprint lengths spanning from 15 to 40 nucleotides, with a prevalent occurrence of footprints measuring 23-25 nucleotides (Supplementary Fig. S9)." The consistency of footprint length distribution was observed, with only a minor distinction noted in Ribo replicate 2 under nitrogen+ conditions, likely due to contamination with longer footprints during size selection. Nevertheless, both Ribo-seq replicates for nitrogen + have a very similar read-count composition.

Comparatively, the predominant 23-25 nucleotide footprints in *M. mazei* were shorter than those commonly observed in eukaryotes (28-30 nucleotides) (Zhang et al., 2018) and the archaeon *Haloflex volcanii* (27-30 nucleotides) (Gelsinger et al., 2020 and Hadjeras et al., 2023), although previous studies have noted a distinct peak at 23-24 nucleotides. Notably, the footprint

distribution in *M. mazei* resembled that reported in the bacterium *E. coli*, characterized by shorter footprints with a peak at 24 nucleotides. Besides we now mention the footprint length of the ribosome (in comparison with Haloferax!) in the revised version (Fig. S9).

Fig. S9: Distribution of Fragment Lengths and Read Density Around Annotated Start Codons.

- (A) Distribution of footprint fragment lengths for each replicate/library, showing a predominance of 23-25nt long fragments.
- (B) Density profiles of 5'-mapped reads around annotated start codons for distinct read lengths in the range of 23-33nt, with reads from all libraries combined.
- (C) Density profiles of 5'-mapped reads around annotated start codons specifically for the most abundant read length of 24nt, with reads from all libraries combined.
- (D) Density profiles of 5'-mapped reads around annotated start codons for each library, including all read lengths in the 23-33nt range.
- (E) Density profiles of 5'-mapped reads around annotated start codons for each library, specifically focusing on the most abundant read length of 24nt.
- (B-E) All profiles demonstrate the highest read density occurring 16nt upstream of the start codon, with a noticeable shift towards position 0 for longer read lengths.

Reviewer #2 (Remarks to the Author):

The paper of Joran et al presents Ribo-seq and RNA-seq analysis of the translome and transcriptome of the archaeon *Methanosarcina mazei* under nitrogen-rich and nitrogen-depleted conditions. On the bases of these data, the authors predict 314 new short ORFs in addition to 93 previously known. They verified that some of these ORFs are indeed expressed in the cell.

Critique

This is a technically sound study, which, however, does not provide major relevations. As the authors accurately summarize at the end of the Introduction section: they provide “a large catalogue of novel *M. mazei* sORFs”. As a resource for *M. mazei* aficionados, this is definitely a very important study. But for a more general scientific community it is less useful because the catalogues of sORFs and small proteins have been described for many bacterial and also archaeal organisms. Because of that, in the opinion of this reviewer, this paper would be an excellent candidate for a more specialized journal.

The paper is clearly written but is unnecessarily long and superfluous. Several sections of the Results section belong to the Methods section. Some parts can be deleted without sacrificing the message. Discussion can be significantly shortened.

Here are just few examples:

II. 390-410. The first two paragraphs of Results are purely technical and belong to the Methods.

II. 442-480. The entire section 3.2 (Insights into protein translation in *M. mazei* through Ribo-seq), including Figure 2, can be eliminated because similar observations (misannotated start codons, truncated or extended genes, etc.) have been described for many genomes.

Too many experimental details spill into Results which, even if counterintuitive, makes reading less informative. For example, Fig. 4A shows the results of sORF prediction by three different methods (!). The paper would benefit if all these go to the Methods section and the readers are simply presented with the most reliable findings. In fact, this entire section (ll. 603-618) can be replaced with just its last sentence: “We can provide a list of 314 small novel proteins with high confidence.”

We certainly agree with the reviewer concerning the long and superfluous manuscript. We have reduced the manuscript thoroughly, transferred methods description from results into Methods (including ll.390-410) and have streamlined the discussion - see also answer to the two other reviewers. We have critically re-examined section 3.2 and shortened it in some places, but have basically retained its content as well as Fig. 2. This is important information for M.mazei and in more general for methanoarchaea, even if this is also the case in other genomes. We agree with the reviewer that there are already numerous studies on the difficulties of genome annotations and correspondingly incorrect or missing annotations for other organisms. However, and importantly, we present the first report on Ribo-seq in combination with biochemical (MS) data and in vivo validation for more than one growth condition. Two conditions and focusing on small proteins has to our knowledge not been reported before (in archaea and bacteria). Retaining Fig. 2 enhances the clarity and visual impact of our findings, offering a concise and illustrative summary that complements the text effectively.

Moreover, while this is not the first study to present proteomics-based evidence for the production of novel sORF translation products, it is the most comprehensive and strongly builds upon previous earlier proteomic analyses. Previous combined MS and Ribo-seq analyses looking to identify the translation products from small ORFs have utilized bottom-up analyses (Hadjeras et al., 2023a and Hadjeras et al., 2023b). This provides inferred identification of small proteins based on the identification of unique peptides originating from the protein. However, while this information is highly valuable, it is surpassed by knowledge of the intact proteoforms that can be gained only by top-down proteomics (TDP) (Cassidy et al., 2023). The direct identification (as opposed to inferred identification gained from bottom-up proteomics analysis) of a number of the novel sORF-encoded proteins via top-down proteomics as full-length or truncated proteoforms is perhaps the strongest evidence available to provide as proof of their existence. This provides us with the knowledge that the novel sORFs are not just transcribed and translated, but that the protein level products (i.e. proteoforms) are present and, at abundances high enough to be detected via TDP. Such insight provides a solid base for future functional assessment of the novel small proteins and provides arguably the highest level of validation towards the results gained by the Ribo-seq analysis.

ll. 603-618 Manual inspection: We agree with the reviewer that the paragraph in question contains too many experimental details and have shortened it in large parts. Nevertheless, we have decided not to shorten it to just one sentence, as we believe that especially the manual inspection with its stringent criteria is a great strength of our work and our dataset and its details

can be important for future studies. Fig. 4A indeed contains a lot of information which is not required in the main manuscript and is now presented in the supplement. Overall, the figures 3, 4, and 5, have been modified and streamlined as also requested by reviewer 1.

The two other reviewers evaluate the manuscript as a great set of new data, 'providing a wealth of informations', 'Given that ribo-seq is still its infancy for archaeal microorganisms, I am convinced that this manuscript is highly valuable for other researchers to further expand this technique in this domain of life'. This evaluation is in direct contrast to the judgment of reviewer 2 'does not provide major revelations'. Moreover, we do believe that the comprehensive data set, especially the list of novel sORFs for *M. mazei* described in this report, is of great interest for several other labs and will play an important role for future functional and genetic elucidations and studies for other archaea as well. Moreover, we have in addition already several running collaborations with other labs in Germany and Europe on those data sets and small proteins. Based on our data sets, which we made available for those collaborators, new insights into at least one large enzyme complex in *M. mazei* has been discovered in the last year by a collaborator.

Reviewer #3 (Remarks to the Author):

In the paper called "Exploring the secret small proteome of *Methanosarcina mazei* through Ribo-seq and peptidomics, uncovering a multitude of dual-function RNAs" by Britta Jordan^{1*}, Muhammad Aammar Tufail^{1*}, Lydia Hadjeras^{2*}, Rick Gelhausen³, Liam Cassidy⁴, Tim Habenicht¹, Miriam Gutt¹, Rolf Backofen³, Andreas Tholey⁴, Cynthia M. Sharma², Ruth A. Schmitz^{1,#}, a Ribo-seq protocol is established for the methanogenic archaeon *Methanosarcina mazei*. This tool provides valuable insights into protein synthesis (e.g. identification of novel coding ORFs and validation of annotated start codons). Ribo-seq was further employed to compare the translome under standard growth conditions and nutrient limitation, with a particular focus on small proteins (< 70 aa). A variety of novel sORFs were predicted and certain candidate sORFs were put forward which possibly play a role in nitrogen metabolism based on phenotypical data. This paper is nicely written and provides a wealth of information. Results are discussed in a pleasant and detailed way with a good number of supporting figures. I appreciate the author's work in confirming the ribo-seq data using mass spec and western blotting. Given that ribo-seq is still its infancy for archaeal microorganisms, I am convinced that this manuscript is highly valuable for other researchers to further expand this technique in this domain of life. My major scientific remark is related to the in vivo analysis of tagged sORFs, where I do not fully support the decisions on episomal overexpression of the sORFs. In addition, I believe that the lay-out and order of the (sub)figures should be improved to get rid of the 'messy' appearance of certain figures.

We thank the reviewer for these valuable comments which will be addressed below. We have optimized the layout of the figures and the order of subfigures, and deleted some subfigures, see comment 2) and 3) below.

Major remarks:

1) "Section 3.5.4 Targeted and untargeted in vivo validation of selected candidates": In this section, it is described that candidate sORFs are selected based on their genomic context (L661) and that these sORFs were episomally expressed in *M. mazei* under control of the constitutive PmcrB promoter and a standard *M. mazei* ribosome binding site and fused to a C-terminal sequential affinity tag (SPA) to assess their in vivo translation (L663-665). I believe this in vivo validation of tagged sORF is a nice contribution to the paper. However, I do not understand the reasoning behind selection of a constitutive PmcrB promoter and a standard *M. mazei* ribosome binding site for driving sORF expression as a first screening in Fig 6A, given that this deviates from the endogenous genomic context of the sORFs (this is not in line with why these candidates were selected in the first place). Based on this general screening, three candidates sORFs were selected to study using their endogenous promoter elements (Fig 6B). However, I believe the latter approach using the native promoter/RBS is more valuable than the analysis using a non-related constitutive promoter. I would furthermore argue that results could be different when using the native promoter elements of the sORF compared to the PmcrB promoter and a standard *M. mazei* ribosome binding site (on plasmid). I would advise to also confirm translation of the other 10 sORFs in the set-up employing their endogenous promoter element. Perhaps some of these other sORFs also display a differential expression in N conditions? In the same line, I wonder why insertion of the epitope tag into the genome was not the preferred option here, as this would be the closest to the natural expression of the ORF (and not overexpress the sORF on plasmid)?

We thank the reviewer for this valuable feedback. Our initial intention was to evaluate whether *M. mazei* is able to produce stable small proteins from the selected sORFs at all as it was done previously (Hadjeras et al., 2023a and Hadjeras et al., 2023b). We however agree that a strong promoter and ribosome binding site in front will most probably lead to protein synthesis. This was the reason why we in addition also looked in three cases where the sORF was under the control of their native promoters and RBS. We are aware and agree with the reviewer that similar analysis for the remaining sORFs only analyzed under the control of a strong promoter might change when expressed under the respective native promoters, as shown for sORF_08. Consequently, we cloned and constructed the majority of sORFs under the native promoter region on a plasmid and transformed the plasmids into *M. mazei* to study their native regulation under N+ and N- conditions and have even included another sORF, namely sORF2, in the analysis and therefore now have a total of 16 selected sORFs for *in vivo* validation. The respective results are now implemented in the revised manuscript (new Fig. 6A). Due to methodological circumstances, not all selected sORFs could be cloned under their native promoter. In summary, 16 sORFs were selected for validation via episomally tagging and western blot. From those 16 sORFs, 10 were cloned under their native promoter fused to a C-terminal SPA-tag. Here, we were able to detect 8 sORFs expressed under their respective native promoter, from which 3 (sORF3, sORF8 and sORF9) showed differential expression

based on the nitrogen availability (see Fig. 6A). As originally described, 13 of the selected sORFs were expressed episomally with a constitutive promoter. Of these 13 sORFs, 10 showed a positive result in the western blot now shown in Fig. S5A. In summary, validation of sORF expression under either the native or a constitutive promoter via epitope tagging and immunoblotting analysis was possible for 13 out of 16 selected sORFs. The missing three sORFs were detected by LC-MS (new Fig. 6B).

The generation of the respective tagged mutants in the genome of *M. mazei* is difficult, not very effective and time-consuming. The transformation has to be done under strict anaerobic conditions and the effectiveness of generating chromosomal mutations is very low even when forced with a selection marker. In addition, *M. mazei* is polyploid, thus generating a mutant with all chromosomes containing the tag takes a long time. We do need however to tag 100% of the chromosomes, otherwise we cannot compare the expression level of different sORFs. Overall, it would have required an exceptionally long time to construct chromosomal mutants for several sORFs (up to a year). Consequently, we decided to use the episomal method but using the native promoters and RBS to be able to finalize the experiments in a decent time and make the Riboseq data set available in a shorter time for other labs.

2) The lay-out of nearly all figures should be improved. The text size in many figures is too small and barely readable, e.g. Fig 3 C-H, Fig 4 B-H, Fig 5E-G. Please also take care to use the same (or at least similar) text fonts, text sizes and color-schemes in different subfigures to avoid the figures looking 'messy' / 'cluttered' (e.g. Fig 3/4/5).

We certainly agree with the reviewer's valuable comment. We thoroughly optimized the figures, increased the font sizes, and took out several text information, and streamlined the colors and font sizes in all figures. Furthermore, we also particularly changed the order, as pointed out by the reviewer.

Fig. 3C-H, Fig. 4B-H, Fig. 5E-G are optimized as requested.

3) In the text, Fig 4A, 4C and 4D are mentioned before Fig 4B. Also for Fig 5 and Fig 7, the order in the text is non-chronological. Please rearrange the labeling of the subfigures.

We agree and have rearranged the labelling in the figures according to the text or we rearranged the text that is now supporting the chronological order of the figures.

Minor remarks:

General:

1) For certain products, the specifications and manufacturer are missing in the M&M section (e.g "M-MLV reverse transcriptase" at L 159, "a high-fidelity DNA polymerase" at L162)

We apologize and have added the missing information in the M&M section in the revised version.

2)What is the size of the generated footprints? Unless I missed this, I could not find this information in the manuscript.

We thank the reviewer for this important point and added the footprint size (23-25 nts) in the results part and in Fig. S9. The sentence now reads: “Our Ribo-seq libraries, obtained under both nitrogen + and - conditions, exhibit a wide range of footprint lengths spanning from 15 to 40 nucleotides, with a prevalent occurrence of footprints measuring 23-25 nucleotides (Fig. S9) Line 847-849”. See also reviewer 1 comment 37.

3)Did the authors provide the log-in details to access the generated Ribo-seq data on <http://www.bioinf.uni-freiburg.de/ribobase> ?

To facilitate a comprehensive evaluation, we have provided the necessary login details and reviewer tokens for accessing the web-based tool discussed in our study, as well as for the online repositories where our data have been deposited. This information has been outlined in the README.pdf and README.md files submitted with the manuscript.

4)There seems to be an issue with the doi-links in the reference list (“<https://doi.org:https://doi.org”>”)

We apologize and took care of that issue.

5)A short section on the (technical) challenges that must be overcome on the implementation of Ribo-seq for anaerobic microorganisms could be beneficial to allow the reader to assess the feasibility of implementing this technique for other archaea (e.g. thermophiles, acidophiles,...).

We thank the reviewer for this valuable comment. We thoroughly condensed the discussion and now implemented this important information in the discussion section:

“Ribo-seq has emerged as a potent method to study protein synthesis, and certainly boasts several advantages over proteomics 98,99. However, it is not advisable to rely solely on Ribo-seq due to the inherent challenges, including complex ribosome behaviours, biases in library preparation, and differential ribosome occupancy 99,100. Consequently, to enhance the reliability of Ribo-seq data, it's essential to combine it with manual curation and filtration of both Ribo-seq and RNA-seq datasets, as well as with validation methods like MS based proteomics and in vivo validation of proteins, as used in recent studies 39,40,48,99.

DeepRibo is prone to a high rate of false positives, which we detect in our results as well, where DeepRibo predicted eight false positive out of 63 translated annotated sORFs and 57 false positives out of 255 predicted novel sORFs (Table S4). Consequently, manual curation of the DeepRibo predictions with stringent cut-off filters based on control proteins (e.g. MM_RS08540) on prediction outputs was performed and provided more confident results in our study.

One major challenge in the data processing is that the effectiveness of bioinformatic workflows like DeepRibo, REPARATION, deltaTE (for differential expression analysis), which are all optimized for bacterial genomes, might be limited on archaeal datasets. However, our optimized workflow using two conditions generated a dataset which provides an important resource for

training these algorithms for archaeal data, suggesting a larger and more comprehensive M. mazei small proteome than currently understood.”

6)L353 “The supernatant was resuspended in LDS-sample buffer (Invitrogen, Darmstadt, Germany) and separated on a NuPAGETM Bis-Tris gel, 12% (Invitrogen, Darmstadt, Germany) at 200 V for 35 min”. How much total protein was loaded on gel? For some blots, this is mentioned in the figure’s legend, but not for all. Please provide this information for all blots.

We apologize to not including this important information in the figure legend and the Material and Method section for all samples. We now add respective information - basically all samples were added to the gels in comparable amounts of 30 µg cell extract protein, which is also true for the newly included western blots regarding the episomally expression of sORFs under their respective native promoter.

Specific:

7)L51-52 “Small proteins in the domain of archaea are underrepresented in recent studies, and only few small proteins are characterized until now 12-14”. The references correspond to small proteins in Euryarchaeotes only. Is it correct to assume that no small proteins have been characterized in other archaeal phyla?

We agree with the reviewer that we only refer to examples of Euryarchaea. However, small proteins are also known for Crenarchaea, for example histone-like proteins in different Sulfolobus strains (Thomm et al., 1982). We have therefore replaced the relevant references with two comprehensive recent reviews of us about small proteins in Archaea, which also include this information (line 55-56).

8)L194 “To ensure the accuracy of the predictions, a rigorous filtering process was performed and manual curation of the predicted ORFs was carried out”. Could you elaborate on which sequences were filtered and how this was performed?

We agree with the reviewer, this is important information. However, we already explained how the data was prefiltered and how the curation was done in the original manuscript (line 203241). The first reviewer even compliments us on the chosen criteria for filtering.

Briefly, following cut-off filters were used: $\text{meanTE} \geq 0.24$ and RNA-seq and Ribo-seq RPKM ≥ 10 (in both replicates) based on the 93 positively labelled as translated annotated sORFs. In addition, novel translated sORF candidates were required to be predicted by DeepRibo with a prediction score > 0 that allows for ORF candidate ranking. The filtered sORFs were then subjected to manual curation with the inspection of the Ribo-seq coverage in a genome browser using these following criteria: i) the shape of the Ribo-seq coverage over the ORF: the evenness of the Ribo-seq coverage was considered and any predicted sORFs exhibiting uneven coverage with peaks in the shape of a plateau, resulting from RNA structures and/or technical biases, we did not consider as a novel sORF; (ii) restriction of Ribo-seq coverage within ORF boundaries (and ribosome footprints excluded from 5'/3'UTRs) was required; (iii) the Ribo-seq signal was generally required to be comparable or higher to the transcriptome signal from the RNA-seq

library and (iv) RNA-seq and Ribo-seq coverage was required to be, generally, at least ten reads per nucleotide normalized (RPKM) by sample size. Where DeepRibo was not able to predict a suitable sORF, we used NCBI ORFfinder (<https://www.ncbi.nlm.nih.gov/orffinder/>) to find the right sORF. In addition, during the manual inspection, some sORFs were found to be translated based on their coverage but did not pass our stringent filtering criteria. Those were added to the list of potentially translated sORFs.

9)L413-418 “Conversely, the sRNA162, a known non-coding regulatory sRNA 87, displayed high cDNA read coverage exclusively in the RNA-seq library (Fig. 1C right panel). This observation further validates its non-coding nature. Furthermore, the read coverages of the 5'- and 3'-UTRs of psmB demonstrated higher levels in the RNA-seq library compared to the Ribo-seq library. This indicates that mRNA regions that are not translated”. How can you explain the presence of (low number of) reads detected in the Ribo-seq for these RNA (especially for sRNA162)? Would this just be background? If so, did you set a threshold for the number of reads below which a non-coding nature could be attributed to? If so, perhaps this could be indicated (as e.g. a dotted line) on the graph?

We thank the reviewer for this valuable comment. Indeed, there are reads in Ribo-seq of small RNAs, which do not contain a sORF - and are not translated either. But this is not background instead this is based on the fact that the structures of small RNAs often contain very stable (loops, other secondary structures). Consequently, the RNA (structure) is not degraded by MNase1 and shows an alleged ribo-footprint which is not the case! This is also the case for riboswitches (often located in 5'- and 3'-UTRs)! Since all sRNAs are different in length and structures and respective stability, it is impossible to determine a cut-off limit for reads obtained by sRNAs or riboswitches.

4)L416 “Furthermore, the read coverages of the 5'- and 3'-UTRs of psmB demonstrated higher levels in the RNA-seq library compared to the Ribo-seq library.” A reference to “(Fig. 1C, left panel)” could be added to make it more clear.

We agree and added the respective reference accordingly (line 414).

5)L483 “*M. mazei* genome has a total of 3,440 annotated ORFs. DeepRibo algorithm predicted 1,566 and 1,430 annotated ORFs under +N and -N conditions, respectively, overall detecting 47% of all annotated ORFs (Fig. 3A).” How does this compare to MS coverage?

The previous bottom-up full proteome analysis of *M. mazei* under -N growth condition allowed for the identification of 2,168 proteins (ca. 65.6%) from the UniProt *M. mazei* database (3,300 entries (63% of 3,440)). Analysis of proteins less than ca. 100 aa residues in length is known to be highly problematic and challenging (Cassidy et al., 2023). Based on the annotated protein sequences present in UniProt, 131 proteins of ≤ 70 aa residues are encoded, of which our bottom-up analysis successfully identified 26 (23%). We would speculate that the difference in what is detected via the two methods may reflect the more transient nature of transcripts and

the longer potential half-life of proteins in a biological setting (albeit that many proteins may undergo a rapid turn-over).

12) L489: “Analysis of the start and stop codon distribution of annotated ORFs (Fig. 3B) shows that *M. mazei* favours the canonical start ATG by 79%. The two mainly used stop codons are TAA (53%) and TGA (42%).”. Are the rarest start and stop codons overrepresented in a particular COG-category? Is there a correlation between start and stop codon usage and a COG-category?

We recognize the importance of this inquiry and have included a supplementary figure (Fig. S3) to illustrate the percentage share of each codon by category. This was achieved by retrieving the CoG category mappings for *M. mazei* from the NCBI CoG database. Subsequently, we calculated the share of each start and stop codon across the categories. Additionally, we introduced a plot that details the number of genes within each category, an essential factor for accurate comparison among categories. The results for both start and stop codons align closely with the distribution observed in the overall annotated genes, with a notable exception. Category X, which encompasses the 'Mobilome: prophages, transposons,' exhibits a significantly higher proportion of TTG start codons (44.9%) compared to other categories.

Fig. S3: Codon Usage Analysis in Clusters of Orthologous Groups (CoGs)

- (A) Percentage share of each start codon within CoG categories.
- (B) Percentage share of each stop codon within CoG categories.
- (C) Size distribution of CoG categories, indicating the number of genes in each category.
- (D) Glossary of CoG category symbols with detailed descriptions. The data was obtained by mapping CoG categories from the NCBI database to our annotated genes.

13) L525 “Among the three most enriched Gene Ontology (GO) terms for biological processes under N limited conditions were ‘nitrogen fixation’, ‘nitrogen cycle metabolic process’, and ‘regulation of nitrogen utilization’ (Fig. 3G).”. I believe it should be worth noting that the response of ‘nitrogen fixation’ and ‘nitrogen cycle metabolic process’ is way more pronounced than

'regulation of nitrogen utilization' (the latter is very comparable to the other groups in the top 10), according to Fig 3G.

We agree and changed the text in the results accordingly (line 520-526).

14) L581: "Under -N, 5 sORFs were found to be upregulated and 11 sORFs downregulated at the RNA level (as seen in Fig. 4F). On the other hand, at the RIBO level, 9 sORFs were observed to be upregulated while 22 sORFs were downregulated (depicted in Fig. 4G), highlighting the complex regulatory mechanisms at work in gene expression." Could you speculate some more on what could be happening here?

We thank the reviewer for this comment. For example, down-regulation is easy to speculate, some of them as indicated in the modified figure version are small proteins from the ribosome, which are down-regulated under N-limitation. This makes sense, since the growth rate of the organism is also very much decreased under nitrogen starvation/N-fixation conditions. The others, e.g., sP36, which is highly upregulated under N-limitation, is already studied and it's function in the regulation of the N-regulation has been discovered - regulating the ammonium transporter AmtB1 in response to an N-upshift (Habenicht et al., 2023). Besides, Moad is important for the Molybdenum pterin cofactor synthesis (cofactor of some of the metabolic enzymes), ferredoxin upregulated under N-limitation is very understandable, since nitrogenase needs a lot of electrons to reduce the N₂ to ammonium. In turn, the electrons are transferred from ferredoxins to the nitrogenase. We now included the regulation of sP36 and the ferredoxin in the main text.

In addition, the data nicely confirm the already observed regulation on the posttranscriptional level e.g. by sRNA154 (see Table 1).

15) L617 "After finalizing the analysis, we can provide a list of 314 small novel proteins with high confidence." A reference to Table S8 at this point would be helpful.

We agree and referred to Table S8 (line 619).

16) You mention at L608 that sORF detection is difficult in long genes. However at L635 "The 314 novel sORFs are encoded in different genomic context (Fig. 5C)[...]". You do compare the genomic regions of the 314 novel sORFs. This makes me wonder whether you could give a prediction/estimation on how many sORFs could potentially still be present in the long genes and you are "missing" here?

We thank the reviewer for this important comment. We are also very curious to determine the number of internal sORFs, e.g. by TIS profiling. However, we do not see any possibility to currently speculate on this number, since the data of canonical Ribo-seq will not allow us to clearly define internal ORF boundaries. Overall, we can now provide a list of 407 small proteins with high confidence, which represent approximately 12% of total ORFs. Considering this number (approx. 12% of all ORFs to be sORFs in *M. mazei* (Jordan et al. 23 - Review)), we would expect a similar number for the internal sORFs.

17) L667 “Out of those 13 candidates, the translation of 10 sORFs was validated in a western blot analysis using a FLAG directed antibody against the SPA-tag (see Fig. 6A). “ A lot of smearing/different sized bands are observed in the blot for sORF_07. How could this be explained?”

We see the point of the reviewer. The predicted molecular mass of the small protein is 6,84 kDa. Thus, we speculate that some of those additional protein bands represent either oligomers of the small protein, while others might represent complexes which contain the detected small protein (his-tagged detection by antibodies) together with a target protein.

18) L728 “We further performed growth experiments with all *M. mazei* mutant strains episomally expressing sORFs. Most of those mutants did not show any phenotype compared to the empty vector control (data not shown).” Under control of which type of promoter/RBS? I would still advise to add the growth curve of all other mutants to the supplementary data of this paper for transparency.

We thank the reviewer for this valuable comment and now present the complete growth analyses of the respective mutants in the supplement (Fig. S5B). The original promoter and RBS of those constructs was based on the non-native promoter (*mcrB*) and RBS. We now also added the growth curves of the new constructs (under control of the native promoter and RBS of the sORFs; see Fig. S4A).

REVIEWERS' COMMENTS

Reviewer #1 (Remarks to the Author):

In my opinion, the authors have adequately reacted to all my comments. The manuscript can be published in the present form.

Reviewer #2 (Remarks to the Author):

The revised version of the paper made me see more positively its novelty and impact. In its revised form, the manuscript is an easier read and the importance of the findings is more evident.

However, I have two outstanding, relatively minor, but yet significant, concern which should be addressed:

l. 411 and Fig. 1C. 'no coverage in the Ribo-seq library' (for the sRNA162 gene) is an overstatement; in fact, there IS Ribo-seq coverage, albeit lower than the coverage in the RNA-seq library, as is evident on the upper part of the right panel of Fig. 1C. Actually, the absolute density of ribosomal footprints for sRNA162 is comparable to that in MM_0694 (on the left panel in the same figure) given the difference in scale of the Y axis. Thus, judging by the figure, the accurate conclusion would be that both RNAs (MM_0694) and sRNA162, are translated, and possibly even generate comparable protein yield, but one mRNA is translated more actively than the other.

Figs. 6A, and S5A. The Western blots lack loading controls. At least a stained gel should be shown, if not in the main figure then at least in the supplementary info file.

The last concern is of more of a cosmetic nature:

LI. 460-461. What do authors mean by "ORF_01 ... is 100% conserved in *M. mazei* using tblastn analysis"? Conserved among various *M. mazei* strains?

Reviewer #3 (Remarks to the Author):

This is the second revision of the manuscript entitled "Exploring the secret small proteome of *Methanosarcina mazei* through Ribo-seq and peptidomics, uncovering a multitude of dual-function RNAs" by Jordan et al.

I appreciate the author's elaborate answers and feedback on my remarks of the original submission and I am very pleased with the work which was put in revising the manuscript accordingly (especially the additional episomal expressions of sORFs under control of their native promoters). The layout of the figures is also much improved. I do not have any further remarks and would like to congratulate the authors with this interesting study.

Reviewer #1 (Remarks to the Author):

In my opinion, the authors have adequately reacted to all my comments. The manuscript can be published in the present form.

Thank you for your constructive comments and your re-evaluation of our manuscript. We greatly appreciate the time and effort you dedicated to reviewing our work. Your feedback has been instrumental in enhancing the quality of our manuscript. We are pleased to hear that the revised manuscript meets your approval.

Reviewer #2 (Remarks to the Author):

The revised version of the paper made me see more positively its novelty and impact. In its revised form, the manuscript is an easier read and the importance of the findings is more evident.

We thank the reviewer for the constructive feedback and appreciation of our work to implement these comments. We are pleased that its novelty and impact are now more evident.

However, I have two outstanding, relatively minor, but yet significant, concern which should be addressed:

I. 411 and Fig. 1C. 'no coverage in the Ribo-seq library' (for the sRNA162 gene) is an overstatement; in fact, there IS Ribo-seq coverage, albeit lower than the coverage in the RNA-seq library, as is evident on the upper part of the right panel of Fig. 1C. Actually, the absolute density of ribosomal footprints for sRNA162 is comparable to that in MM_0694 (on the left panel in the same figure) given the difference in scale of the Y axis. Thus, judging by the figure, the accurate conclusion would be that both RNAs (MM_0694) and sRNA162, are translated, and possibly even generate comparable protein yield, but one mRNA is translated more actively than the other.

We already explained that ncRNA162, the regulatory non-coding RNA, looks like as being translated. However, since the regulatory function of sRNA162 has been confirmed as well as the structure determined and both is published, we are very confident that those reads in the Ribo-seq approach reflect the high stability of secondary structures of the sRNA162 particularly in the 3' and 5' regions, which keep the RNases (MNase) away from degrading. Here is our previous comment to reviewer 3 minor remarks #9.

"We thank the reviewer for this valuable comment. Indeed, there are reads in Ribo-seq of small RNAs, which do not contain a sORF - and are not translated either. But this is not background instead this is based on the fact that the structures of small RNAs often contain very stable

(loops, other secondary structures). Consequently, the RNA (structure) is not degraded by MNase1 and shows an alleged ribo-footprint which is not the case! This is also the case for riboswitches (often located in 5'- and 3'-UTRs)! Since all sRNAs are different in length and structures and respective stability, it is impossible to determine a cut-off limit for reads obtained by sRNAs or riboswitches.”

Figs. 6A, and S5A. The Western blots lack loading controls. At least a stained gel should be shown, if not in the main figure then at least in the supplementary info file.

We thank the reviewer for this important note and now included the loading control gels, which were performed in parallel and stained with Coomassie blue for all Western blots in supplemental Fig. S6.

The last concern is of more of a cosmetic nature:

LI. 460-461. What do authors mean by “ORF_01 ... is 100% conserved in *M. mazei* using tblastn analysis”? Conserved among various *M. mazei* strains?

We agree with the reviewer that this sentence is somewhat unclear. ORF_01 is indeed conserved at DNA level to 100% among various *M. mazei* strains (eg. Gö1, C16, S-6, LYC and even more). We have amended this sentence by adding the word “various” so that this result is now more comprehensible.

Reviewer #3 (Remarks to the Author):

This is the second revision of the manuscript entitled "Exploring the secret small proteome of *Methanosarcina mazei* through Ribo-seq and peptidomics, uncovering a multitude of dual-function RNAs" by Jordan et al.

I appreciate the author's elaborate answers and feedback on my remarks of the original submission and I am very pleased with the work which was put in revising the manuscript accordingly (especially the additional episomal expressions of sORFs under control of their native promoters). The layout of the figures is also much improved. I do not have any further remarks and would like to congratulate the authors with this interesting study.

We sincerely thank you for your thorough review and the positive feedback on the second revision of our manuscript. We are grateful for your appreciation of the efforts we made to address your initial remarks. Your insightful comments were crucial in guiding these

enhancements and in refining our study and manuscript overall. Thank you once again for your valuable contributions to our work.